# PCGF1-PRC1 links chromatin repression with DNA replication during hematopoietic cell lineage commitment

Junichiro Takano[1,2,3], Shinsuke Ito[2], Yixing Dong[2], Jafar Sharif [2],
Yaeko Nakajima-Takagi[4], Taichi Umeyama[5], Yong-Woon Han [6], Kyoichi Isono[2,7],
Takashi Kondo [2], Yusuke Iizuka[2], Tomohiro Miyai [2], Yoko Koseki[2],
Mika Ikegaya[1,2], Mizuki Sakihara[8], Manabu Nakayama[9], Osamu Ohara[9],
Yoshinori Hasegawa[9], Kosuke Hashimoto[10,11], Erik Arner [12], Robert J. Klose [13],
Atsushi Iwama[4], Haruhiko Koseki [2,3] ✉ & Tomokatsu Ikawa[1,8] ✉

Polycomb group proteins (PcG), polycomb repressive complexes 1 and 2 (PRC1 and 2), repress lineage inappropriate genes during development to maintain proper cellular identities. It has been recognized that PRC1 localizes at the replication fork, however, the precise functions of PRC1 during DNA replication are elusive. Here, we reveal that a variant PRC1 containing PCGF1 (PCGF1-PRC1) prevents overloading of activators and chromatin remodeling factors on nascent DNA and thereby mediates proper deposition of nucleosomes and correct downstream chromatin configurations in hematopoietic stem and progenitor cells (HSPCs). This function of PCGF1-PRC1 in turn facilitates PRC2-mediated repression of target genes such as *Hmga2* and restricts premature myeloid differentiation. PCGF1-PRC1, therefore, maintains the differentiation potential of HSPCs by linking proper nucleosome configuration at the replication fork with PcG-mediated gene silencing to ensure life-long hematopoiesis.

Polycomb group (PcG) proteins are epigenetic modifiers that play a role to maintain the cellular identities of hematopoietic cells during their differentiation by optimizing the expression of developmental- and differentiation-related genes[1-4]. Chromatin repression by PcG factors is mediated by at least two distinct PcG complexes, namely, Polycomb repressive complexes-1 and −2 (PRC1 and 2)[5]. PRC1 mediates mono-ubiquitination of histone H2A at lysine 119 (H2AK119ub1) via the activities of the E3 ubiquitin ligases RING1A and RING1B[6]. On the other hand, PRC2 mediates trimethylation of histone H3 at lysine 27 (H3K27me3) via the activities of the histone methyltransferases EZH1/ 2, in association with PRC2 core factors EED and SUZ12[7,8]. To enable timely activation or inactivation of PcG target genes during

[1]Laboratory for Immune Regeneration, RIKEN Center for Integrative Medical Sciences (RIKEN-IMS), Yokohama, Kanagawa, Japan. [2]Laboratory for Developmental Genetics, RIKEN Center for Integrative Medical Sciences, Yokohama, Kanagawa, Japan. [3]Department of Cellular and Molecular Medicine, Graduate School of Medical and Pharmaceutical Sciences, Chiba University, Chiba, Japan. [4]Division of Stem Cell and Molecular Medicine, Center for Stem Cell Biology and Regenerative Medicine, The Institute of Medical Science, University of Tokyo, Tokyo, Japan. [5]Laboratory for Microbiome Sciences, RIKEN-IMS, Yokohama, Kanagawa, Japan. [6]Laboratory for Integrative Genomics, RIKEN-IMS, Yokohama, Kanagawa, Japan. [7]Laboratory Animal Center, Wakayama Medical University, Wakayama, Japan. [8]Division of Immunology and Allergy, Research Institute for Biomedical Sciences, Tokyo University of Science, Noda, Chiba, Japan. [9]Chromosome Engineering Team, Department of Technology Development, Kazusa DNA Research Institute, Kisarazu, Japan. [10]Laboratory of Computational Biology, Institute for Protein Research, Osaka University Osaka, Japan. [11]Laboratory for Transcriptome Technology, RIKEN-IMS, Yokohama, Kanagawa, Japan. [12]Laboratory for Applied Regulatory Genomics Network Analysis, RIKEN-IMS, Yokohama, Kanagawa, Japan. [13]Department of Biochemistry, University of Oxford, Oxford, UK. ✉e-mail: haruhiko.koseki@riken.jp; ikawa@rs.tus.ac.jp

development and differentiation, PcG proteins are in general counteracted by gene activation programs mediated by the Trithorax group (TrxG) of proteins[9–12]. TrxG complexes also consist of two main subgroups, the COMPASS (complex of proteins associated with Set1) and the SWI/SNF (switching/sucrose non-fermenting) chromatin remodelers, which exhibit mutually overlapping functions. COMPASS regulates mono-, di- and trimethylation of histone H3 lysine 4 (H3K4) and thereby facilitates active transcription, while SWI/SNF regulates ATP-dependent chromatin remodeling mediated by BRG1, which is an ATPase subunit from the SNF2 family[9]. As TrxG factors also play key roles in normal as well as malignant hematopoiesis[9,13–17], TrxG could contribute to the regulation of PRC-dependent chromatin repression to facilitate proper differentiation of hematopoietic cells.

The hematopoietic system originates from hematopoietic stem cells (HSCs)[18], which give rise to multipotent hematopoietic progenitor cells (HPCs) during early hematopoiesis. These HPCs, in turn, generate large pools of committed progenitor/precursor cells[19]. Furthermore, HSCs and HPCs undergo multiple divisions during which some of these cells acquire lineage-specific programs and proceed to commitment. Importantly, to maintain the correct differentiation potential of hematopoietic stem/progenitor cells (HSPCs), PRC-dependent chromatin repression must be restored after each cycle of DNA/chromatin replication during the S-phase[20]. Indeed, previous studies revealed that PRC1 is associated with the replication fork during DNA/chromatin replication[21–23]. However, the precise mechanism(s) by which PRC1 maintains a repressive chromatin state beyond successive DNA/chromatin replication is not well understood.

PRC1 forms at least six different sub-complexes, incorporating six alternative PCGF (Polycomb group ring fingers, PCGF 1 to 6) proteins[24], and it is expected that one or more of these sub-complexes should be involved in mediating chromatin repression. Consistent with this notion, previous studies have linked a variant PRC1 complex containing PCGF1 (PCGF1-PRC1), RING1A or B, KDM2B (lysine demethylase 2B), and BCOR (BCL6 corepressor) with PcG-mediated gene repression in embryonic stem cells (ESCs)[25,26]. Mechanistically, KDM2B recognizes unmethylated CpG islands via its CXXC-motif and recruits PCGF1-PRC1 to target genes. Furthermore, H2AK119ub1, mediated by PCGF1-PRC1, facilitates recruitment of PRC2 and, in turn, deposition of H3K27me3[5,25]. Local H3K27me3 recruits canonical PRC1 (cPRC1) that contains PCGF2 (MEL18) or PCGF4 (BMI1). The cPRC1 mediates condensation of chromatin[27] and establishes transcriptionally repressive states. This interplay between PCGF1-PRC1 and cPRC1 indicates that PCGF1-PRC1 should function upstream of cPRC1.

Intriguingly, however, the situation appears to be different in hematopoiesis than in ESCs. Indeed, it has been shown that PCGF4 (BMI1), a cPRC1 component, is essential in the maintenance of the self-renewal capacity of HSCs[28,29] by suppressing the cell cycle regulator INK4A/ARF[30] and inhibiting premature activation of B cell master regulators such as PAX5 and EBF1[4]. Interestingly, depletion of PCGF1-PRC1 components, such as BCOR[31,32] or KDM2B[33,34], does not affect the self-renewal capacity of HSCs, but instead skews the differentiation of HPCs into the myeloid lineage. In addition, BCOR suppresses leukemic transformation of HSPCs[31,32], whereas PCGF4 facilitates maintenance of leukemic stem cells[29,35]. Therefore, the interplay between PCGF1-PRC1 and cPRC1 observed in ESCs may not occur during normal hematopoiesis, indicating the presence of yet unknown mechanisms by which PCGF1-PRC1 mediates gene repression in hematopoietic cells[36].

It is also important to note that the findings based on perturbation of *Bcor* or *Kdm2b* may not wholly reflect the function of PCGF1-PRC1, as BCOR and KDM2B are known to form complexes, not only with PCGF1-PRC1 but also with non-PcG proteins such as BCL6 and SKP1, respectively. In contrast, PCGF1 is preferentially incorporated into the PCGF1-PRC1 complex and requires RING1A/B for target binding in ESCs[37]. Consistent with this observation, it was reported that PCGF1 co-purified mainly with PcG-related factors[24]. We, therefore, generated a conditional *Pcgf1* mutant allele (*Pcgf1*-cKO) to directly evaluate the role of PCGF1-PRC1 during early hematopoiesis (Supplementary Fig. 1c). We found that *Pcgf1* deletion promotes precocious myeloid differentiation of HSPCs and suppresses lymphoid potential. Consistent with this observation, PCGF1-PRC1 and PRC2 bind to and down-regulate myeloid genes. Intriguingly, although PCGF1-PRC1 facilitates H3K27me3 deposition at target genes, binding of PRC2 components is unaffected by *Pcgf1* depletion, indicating that PCGF1 suppresses target genes via other mechanisms. Supporting this idea, our proteomics analysis revealed that PCGF1 localizes at the replication fork together with proteins associated with the replication machinery and prevents excessive loading of the BRG1 chromatin remodeler on nascent DNA. PCGF1-PRC1, thereby, plays a role to deposit nucleosomes immediately after the passage of a replication fork and this function of PCGF1-PRC1 is required for inheritance of proper chromatin conformation following DNA replication. Given that PRC2-mediated H3K27me3 also depends on proper nucleosome configuration[38,39], this function of PCGF1-PRC1 could, in turn, influence the efficiency of H3K27me3 deposition. Taken together, our studies reveal a role of PCGF1-PRC1 to repress premature differentiation of HSPCs into the myeloid lineage by regulating PRC2-mediated H3K27me3 via a replication-coupled mechanism.

## Results

### PCGF1-PRC1 represses precocious activation of the myeloid program in HSPCs

As described above, PRC1 plays critical roles for regulation of HSPC homeostasis[1]. Consistent with this notion, *Pcgf1* is expressed in HSPCs and hematopoietic cells at different HSPC stages. We found that the highest level of *Pcgf1* is observed in the multipotent progenitor (MPP) 1 stage (Supplementary Fig. 1a, b)[40]. To determine the role of PCGF1 during hematopoiesis, we took advantage of a bone marrow (BM) transplantation system, in which BM cells of Cre-ERT2:*Pcgf1*fl/fl (PCGF-deficient) or Cre-ERT2 (control) mice are transplanted into a recipient mouse after lethal irradiation. *Pcgf1* deletion in donor cells (Cre-ERT2:*Pcgf1*fl/fl) was induced by intraperitoneal injection of tamoxifen 4 weeks after transplantation (Fig. 1a) and the mice were analyzed at 8 weeks after tamoxifen treatment. We confirmed by genomic PCR that exons 2-7 of the *Pcgf1* gene were efficiently deleted in hematopoietic cells in the tamoxifen treated mice (Fig. 1b, Supplementary Fig. 1c). As expected, *Pcgf1* transcript level was markedly reduced in lymphoid-primed multipotent progenitors (LMPPs) (Supplementary Fig. 1d).

We noted that the total number of BM cells and splenocytes in the recipient mice reconstituted with *Pcgf1*-KO BM was decreased, while the number of thymocytes was unaffected (Supplementary Fig. 1e,f). Strikingly, *Pcgf1*-deficient BM cells failed to efficiently generate the B cell lineage (Fig. 1c, Supplementary Fig. 1g), while the mature myeloid cell population in the BM was expanded (Fig. 1c, Supplementary Fig. 1g). Furthermore, analysis of HSPCs in *Pcgf1*-KO BM revealed significant reduction in both frequency and number of MPPs, LMPPs[41] and common lymphoid progenitors (CLPs), all of which possess lymphoid potential. The number of HSCs, however, barely changed. In contrast, granulocyte monocyte progenitors (GMPs) were significantly increased (Fig. 1d, Supplementary Fig. 1h, i). Collectively, these results indicate that differentiation of *Pcgf1*-KO HSPCs is biased toward the myeloid lineage.

To further elucidate the changes in the differentiation dynamics of HSPCs in the *Pcgf1*-KO, we performed single cell (sc) RNA-sequencing using lineage-marker negative (Lin⁻)Sca-1⁺c-kit⁺ (LSK) cells from BM of Cre-ERT2 (control) or Cre-ERT2:*Pcgf1*fl/fl mice (Fig. 1a). We used 3155 control and 7854 *Pcgf1*-KO single cells and identified nine major clusters (C) based on dimension reduction by UMAP (Uniform Manifold Approximation and Projection)[42] (Fig. 1e). Functional annotation of respective UMAP clusters was performed by comparing our

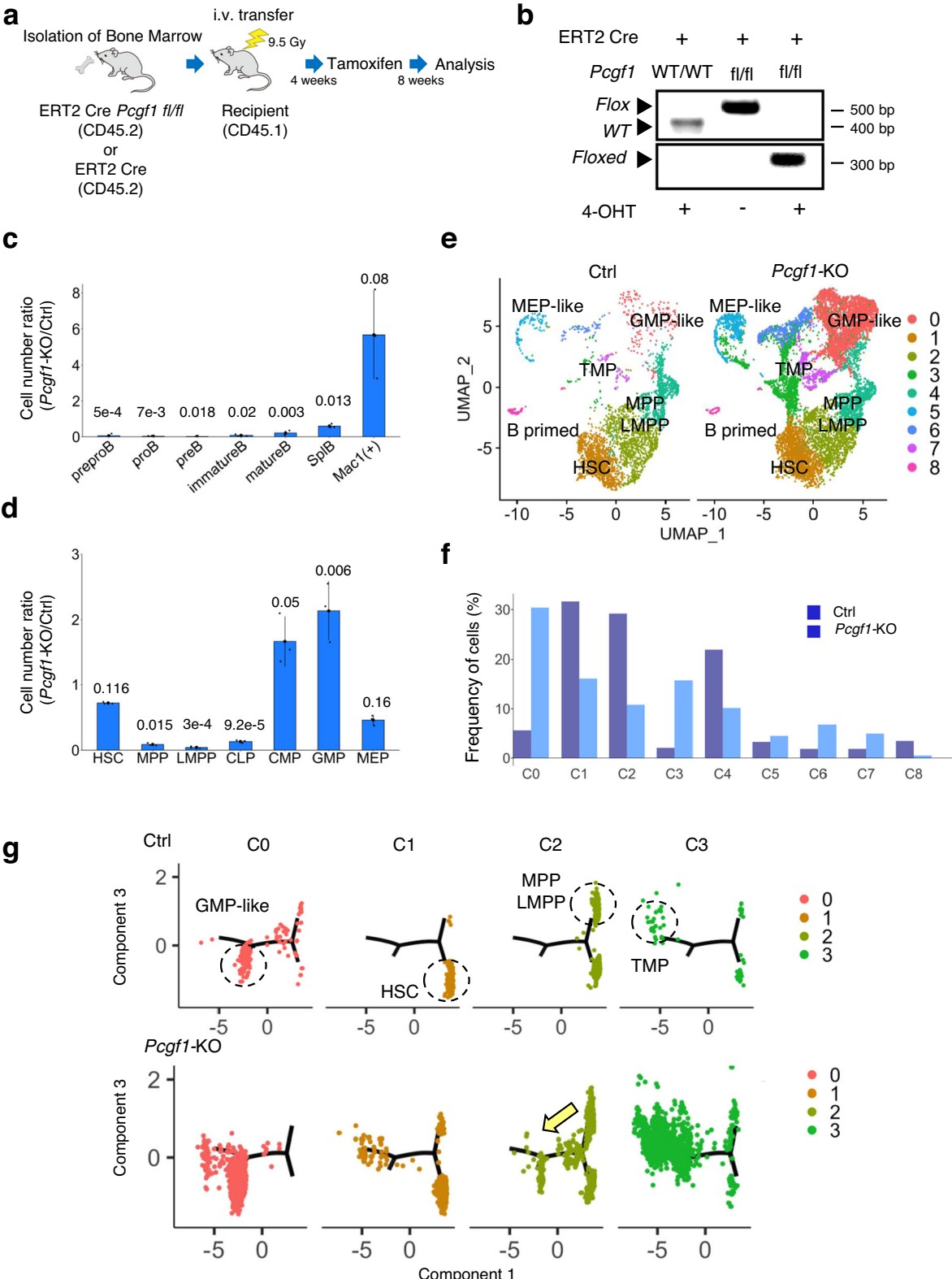

results with previously reported HSPC gene expression profiles[43] (Supplementary Fig. 1j). These analyses enabled us to assign C0, C1, C2/4, and C5 to GMP-like, HSCs, MPP/LMPP, and MEP-like cells, respectively (Fig. 1e). We could not annotate C3, C6, C7, and C8 clusters by this comparison. However, as analysis of marker genes revealed strong expression of myeloid genes in C3, C6, and C7 (Supplementary Fig. 1k), we regarded them as transitional myeloid progenitors (abbreviated TMP hereafter). In addition, the C8 cluster could be designated as B

lineage progenitors according its gene expression pattern. We could annotate more than 80% of the control LSK cells as HSC or MPP/LMPP (Fig. 1e, f), as expected. Interestingly, while we could barely detect GMP-like or MEP-like cells in the control, *Pcgf1*-ablated cells showed an increase in GMP-like cells (C0), and TMPs (C3, C6 and C7) (Fig. 1e, f). In contrast, the frequency of HSC (C1), MPP/LMPP (C2 and C4) and B-primed cells (C8) decreased in *Pcgf1*-KO LSK cells (Fig. 1f).

**Fig. 1 | PCGF1 represses ectopic myeloid differentiation in early hematopoiesis.**
**a** Schematic representation of the experimental procedure. Briefly, 1×10⁶ bone marrow (BM) cells from ERT2-Cre;*Pcgf1*fl/fl or ERT2-Cre mice, both on a CD45.2 background, were transplanted into 9.5 Gy irradiated CD45.1 recipient mice. Bone marrow chimeras were treated with Tamoxifen at 4 weeks after transplantation and subjected to various analyses at 8 weeks after Tamoxifen administration. **b** Induced deletion of *Pcgf1* exons 2 to 7 in LSKs of ERT2 Cre *Pcgf1*fl/fl mice by 4-OHT treatment. Floxed, floxed *Pcgf1* allele. **c** Defects in B lineage development in BM (preproB – mature B) and spleen (SplB) of mice reconstituted with *Pcgf1*-KO cells. On the other hand, myeloid cells [Mac 1 (+)] were expanded. **d** Marked reduction of lymphoid-primed multipotent progenitor cells and increase of granulocyte macrophage progenitors in BM reconstituted with *Pcgf1*-KO cells. HSC: hematopoietic stem cells, MPP: multipotent progenitors, LMPP: lymphoid-primed multipotent progenitors, CMP: common myeloid progenitors, GMP: granulocyte macrophage progenitors, MEP: megakaryocyte-erythroid progenitors, and CLP: common lymphoid progenitors. Procedures to identify respective fractions in **c**, **d** are shown in Supplementary Fig. 1g, h, i. Bar graphs in **c** and **d** represent mean ± standard deviation (SD) of rations of cell numbers (*Pcgf1*-KO/Ctrl) derived from independent biological triplicates. The numbers on the graph are *p*-values between the control and *Pcgf1*-KO calculated with the Welch's two-sided *t* test. **e** Changes in differentiation paths of LSK (Lin⁻c-Kit⁺Sca-1⁺) cells in *Pcgf1*-KO revealed by single cell RNA-seq (scRNA-seq) analysis. Control (Ctrl) and *Pcgf1*-KO LSK cells were stratified into 9 clusters as illustrated by the UMAP plot. By comparing these data with publicly available data for HSPCs, cellular properties of respective clusters were annotated. TMP: Transitional myeloid progenitors. **f** Bar graphs comparing the frequency of cells in each cluster shown in **e** between control (Ctrl) and *Pcgf1*-KO. **g** An inferred pseudo-time trajectory of each cluster defined in **e**. GMP-like cells, HSCs, MPPs/LMPPs and TMPs are expected to be enriched on the trajectory as indicated by dotted circles in the control. Premature differentiation paths of HSPCs toward TMPs and GMP-like cells are expected to be activated in *Pcgf1*-KO.

Next, we inferred pseudo-time differentiation trajectories of LSK cells by using Monocle[44], and assigned respective clusters on the trajectories to infer LSK differentiation paths (Fig. 1g, Supplementary Fig. 1k, l). In control cells, the C1 cluster was allocated to the end of lower right branch of a tree-like trajectory, while the C2 cluster was allocated to the lower and upper ends of the right branches. In contrast, cells in the C0 or the C3 cluster were allocated to the lower or upper left ends, respectively. As HSCs were enriched in C1, we assigned the right lower area occupied by C1 cells as the root state. From the root state, we inferred the directions for MPP/LMPPs, GMP-like and TMPs, radiating to the right upper, left lower, and left upper directions, respectively. In the *Pcgf1*-KO, a considerable fraction of cells in the C1, C2 or C4 clusters were positioned at the area designated for GMP-like cells and TMPs. Furthermore, we found a sequential lineage differentiation path from MPP/LMPP to GMP-like cells in C2 and C4 that was not detected in control cells. These results indicated that HSCs and MPP/LMPPs ectopically and/or prematurely acquire myeloid properties in the *Pcgf1*-KO. Taken together, we conclude that PCGF1 suppresses premature activation of myeloid programs in HSPCs to avoid ectopic differentiation of TMPs and reciprocal exhaustion of lymphoid cells.

## Establishment of *Pcgf1*-KO multipotent progenitor cell lines

Next, we sought to clarify the molecular mechanism by which PCGF1 regulates HSPC cell fate. However, the heterogeneity of HPCs and their propensity to rapidly differentiate into lineage committed progenitors make epigenetic analysis problematic in these cells. To circumvent this problem, we captured a subset of HPCs in ex vivo culture, following a method that we have described previously[45]. In brief, we isolated LSKs from mice that harbor a conditional mutation in *Pcgf1* or in other PRC1 components[32,46,47]. We then expressed Id3 (inhibitor of DNA binding 3), which suppresses HPC lineage commitment by inhibiting the master regulator E2A (transcription factor 3: TCF3)[45,48]. To establish immortalized cell lines, we cultured these LSKs under B cell differentiation conditions by supplementing the media with SCF, IL-7 and Flt3-L and using TSt-4 stromal cells as feeders. Of note, such Id3-induced HPCs (hereafter called IdHPCs) possess the potential to generate T, B and myeloid cells in vivo[45], similar to LMPPs, and have a surface phenotype identical to LMPPs (Flt3⁺/CD34⁺, Supplementary Fig. 2a). These cells, therefore, can be utilized as a substitute for LMPPs[45,48].

## PCGF1-PRC1 down-regulates a group of genes related to myeloid differentiation

We found that PCGF1 was efficiently depleted in IdHPCs after 4 days of 4-OHT treatment (Supplementary Fig. 2b) and that *Pcgf1-KO* IdHPCs showed a modest but significant decrease in proliferation capacity (Supplementary Fig. 2c). *Pcgf1* depletion, however, did not perturb the levels of RING1B, H2AK119ub1, SUZ12, or H3K27me3 (Supplementary Fig. 2b), or of surface markers of IdHPCs (CD34⁺ Flt3⁺LSK)

(Supplementary Fig. 2a). Next, we explored the genome-wide distribution of PCGF1, RING1B and H3K27me3 in IdHPCs. As there is no commercially available chromatin immunoprecipitation (ChIP)-grade antibody to PCGF1, we expressed an exogenous 3xFLAG-tagged PCGF1 in *Pcgf1*-KO IdHPCs and performed ChIP analysis using an anti-FLAG antibody. 3xFLAG-PCGF1 was predominantly enriched at CGI-containing gene promoters (Supplementary Fig. 2d). In particular, we identified 1574 genes with PCGF1 peaks around their promoter regions (hereafter described as PCGF1 target genes)(Fig. 2a). Furthermore, as we were aware that assays based on exogenous genes are sometimes problematic, we generated IdHPCs that harbor endogenous TY1-tagged PCGF1 (Supplementary Fig. 2e, f) and conducted ChIP-seq for TY1-tagged PCGF1. In this case, the binding pattern of endogenous TY1-tagged PCGF1 was very similar to exogenous 3xFLAG-tagged PCGF1 (Fig. 2b). Importantly, 82% of the PCGF1 target genes were co-occupied by RING1B, representing the targets of the PCGF1-PRC1 complex (Fig. 2a). Intriguingly, when we examined the overlap of PCGF1-PRC1 bound genes with H3K27me3, we found only a 37% overlap (Fig. 2a). This uncoupling of the majority of PCGF1-PRC1 bound genes from H3K27me3 (RING1B⁺H3K27me3⁻) in IdHPCs represents a marked contrast with ESCs, in which the RING1B⁺H3K27me3⁻ fraction is barely detected (Supplementary Fig. 2g)[25,47].

We sub-divided the genes bound by PCGF1-PRC1 into two groups, namely, PCGF1⁺RING1B⁺H3K27me3⁺ (Cluster 1, or C1), and PCGF1⁺RING1B⁺H3K27me3⁻ (Cluster 2, or C2) and further classified the rest of the genes as Cluster 3 (C3): PCGF1ˡᵒRING1BˡᵒH3K27me3⁻CpG⁺ and Cluster 4 (C4): PCGF1⁻RING1B⁻H3K27me3⁻ CpG⁻ based on local levels of the CpG signal (Fig. 2b). To compare the chromatin features between C1 and C2 genes, we examined the distribution of H2AK119ub1, SUZ12 (a PRC2 component), PHC2 (a canonical PRC1 component), H3K27ac, and RNA polymerase II (RNAPII) (Fig. 2b). H3K27me3, SUZ12 and PHC2 were enriched only at C1 genes, while PCGF1, RING1B, and H2AK119ub1 were enriched at both C1 and C2 genes. In contrast, active chromatin marks such as H3K27ac and RNAPII were enriched in C2 (Fig. 2b). These results indicated that C1 genes could be the main targets of PcG-mediated gene repression. Indeed, by performing RNA-seq, we observed that the expression levels of C1 genes were significantly lower than those of C2, C3 and C4 (Fig. 2c).

Upon ablation of *Pcgf1* in IdHPCs, 323 genes were up-regulated, and 72 were down-regulated (Supplementary Fig. 2i). As expected, up-regulated genes were enriched with C1 (Fig. 2c, Supplementary Fig. 2i, j). To reveal the functional characteristics of C1 and C2 genes, we performed Gene ontology (GO) analysis and found enrichment of developmental-, stem cell- and myeloid-related terms in both Clusters (Fig. 2d). In contrast, lymphoid- and B cell-related genes were enriched only in Cluster 2. However, myeloid- and stemness-related genes included in C1 were less abundantly expressed than those included in C2 (Fig. 2e). The same group of genes was also up-regulated upon

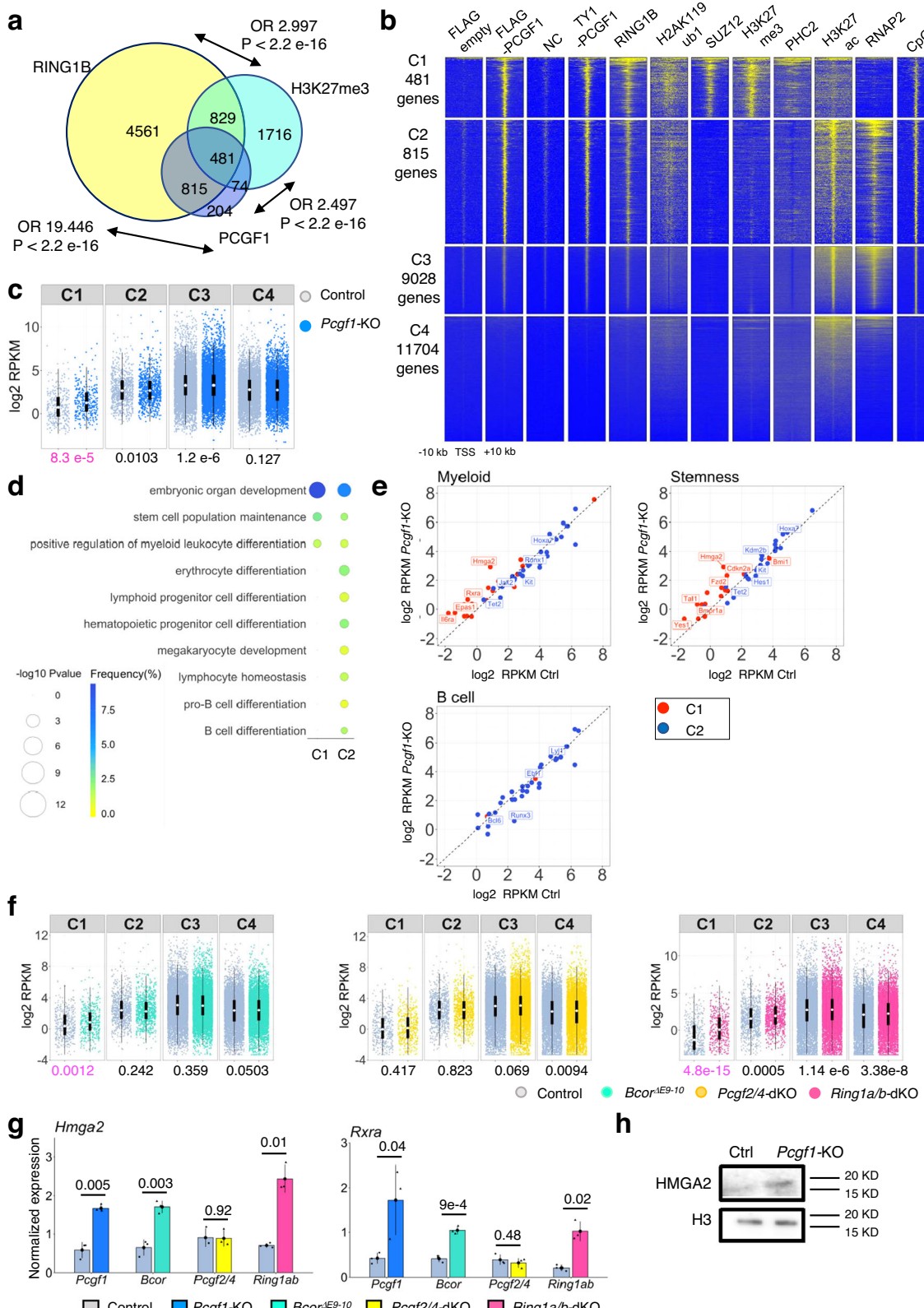

PCGF1 depletion. In contrast, B cell-related genes, enriched in C2, were not sensitive to PCGF1 depletion. Collectively, these data revealed a primary role of PCGF1 is to repress myeloid- and stemness-related genes in IdHPCs.

To further explore if this regulation of C1 genes by PCGF1 reflects a function of PCGF1-PRC1, we analyzed the transcription profile of IdHPCs in which *Bcor*[32] (a representative component of PCGF1-PRC1),

*Pcgf2/4*[46] (central component of cPRC1) or *Ring1a/b*[47] (a universal component of PRC1) were disrupted (Supplementary Fig. 2k, l, m). In the *Bcor*[ΔE9-10] mutant the PCGF1-PRC1 sub-complex is perturbed, while in the *Pcgf2/4*-dKO mutant the cPRC1 sub-complexes are perturbed, and finally in the *Ring1a/b*-dKO mutant all PRC1 sub-complexes are perturbed. A *Bcor* loss of function mutation in which exons 9 to 10 are deleted (*Bcor*[ΔE9-10]), thereby binding to PCGF1 is disrupted, as well as

**Fig. 2 | Repressive role of PCGF1 for a subset of PCGF1-PRC1 target genes in IdHPCs. a** Considerable overlap of genes bound by exogenous PCGF1, RING1B and H3K27me3 in IdHPCs. Odds ratio (OR) for each overlap is shown with the *p*-value estimated by Fisher's exact test (two-sided) without adjustment of multiple comparisons. **b** Heatmap of ChIP-seq signals for indicated antibodies and distribution of unmethylated CpG dyads (CpG) across TSS (±10 kb) in IdHPCs. Representative data of two biologically independent experiments are shown except for the negative control for FLAG-PCGF1, which was obtained from a single experiment. **c** Box plot of gene expression in each cluster in control and *Pcgf1*-KO IdHPCs. Data in graphs represent average for two biologically independent experiments. The center white circle indicates a median value and the boxes indicate 25th to 75th percentile. *p*-values calculated with the Wilcoxon signed rank test are shown. RPKM: Reads Per Kilobase of exon per Million. Low abundance genes (CPM < 1 across all samples) were excluded. **d** Gene ontology (GO) analysis for PCGF1 target genes. Selected GO terms for C1 and C2 are shown. Odds ratio and *p*-values are estimated

by clusterprofiler[75]. **e** Gene expression changes of "Myeloid", "Stemness", and "B cell" related genes in *Pcgf1*-KO IdHPCs. Data in graphs represent mean for two biologically independent experiments. Red and blue dots denote C1 and C2 genes, respectively. **f** Box plot for gene expression in each cluster in *Bcor*^ΔE9-10 (green), *Pcgf2/4*-dKO (yellow), and *Ring1a/b*-dKO (red) IdHPCs with respective controls (light blue). Data in graphs represent mean for two biologically independent experiments. The center circle indicates a median value and the boxes indicate 25th to 75th percentile. *p*-values calculated with the Wilcoxon signed rank test are shown. **g** Up-regulation of *Hmga2* and *Rxra* in *Pcgf1*-KO, *Bcor*^ΔE9-10 and *Ring1a/b*-dKO, but not in *Pcgf2/4*-dKO as shown by RT-qPCR analysis. Data represent mean ± SD of three independent experiments. *p*-values between the control and respective mutants calculated with the Welch's two-sided *t* test are shown. **h** Up-regulation of HMGA2 in *Pcgf1*-KO IdHPCs shown by immunoblot. Data shown in the blot are representative of two independent experiments.

deletion of *Pcgf2* and *4* were achieved by 4 days of 4-OHT treatment, while *Ring1b* was depleted in *Ring1a-/-* IdHPCs by 2 days of 4-OHT treatment. As expected, similar up-regulation of C1 genes was also observed in *Bcor*^ΔE9-10 and *Ring1a/b*-dKO IdHPCs, to a higher level in the *Ring1a/b*-dKO (Fig. 2f, Supplementary Fig. 2n). This up-regulation, however, was not observed in *Pcgf2/4*-dKO IdHPCs (Fig. 2f, Supplementary Fig. 2n), indicating that PCGF1 regulates down-regulation of C1 genes as a part of the PCGF1-PRC1 complex. We also confirmed reactivation of representative C1 genes such as *Hmga2* and *Rxra* in *Pcgf1*-KO, *Bcor*^ΔE9-10 and *Ring1A/B*-dKO but not *Pcgf2/4*-dKO IdHPCs by RT-qPCR analysis (Fig. 2g). Moreover, immunoblotting (IB) analysis revealed that the HMGA2 protein level was significantly elevated in *Pcgf1-KO* IdHPCs (Fig. 2h). Taken together, these results show that PCGF1-PRC1 contributes to down-regulation of the C1 genes in IdHPCs. In contrast, canonical PCGF2/4-PRC1 are dispensable for this process.

## PCGF1-PRC1 down-regulates C1 genes by an H3K27me3-dependent mechanism

We next examined how PCGF1-PRC1 regulates the expression of C1 genes in IdHPCs. Given that the PCGF1-PRC1 complex mediates gene silencing via chromatin-dependent mechanisms, and that the KDM2B/H2AK119ub1/JARID2 axis plays a critical role in this process[25], we performed ChIP-seq to elucidate the distribution of KDM2B, BCOR, RING1B, H2AK119ub1, JARID2, SUZ12, H3K27me3, and PHC2 (a key component of cPRC1) in control or *Pcgf1*-KO (Fig. 3a, b). Surprisingly, although H3K27me3 levels at C1 genes were considerably decreased in *Pcgf1*-KO IdHPCs, KDM2B, BCOR, RING1B, H2AK119ub1, JARID2, SUZ12, and PHC2 levels remained unaltered (Fig. 3a, b). ChIP-qPCR analysis at selected C1 (*Ink4a, Hmga2, Rxra*) and C2 (*Runx3, Rb1*) genes also supported this conclusion (Fig. 3c). Therefore, reduction of H3K27me3 in C1 genes does not accompany a decrease of H2AK119ub1, PRC2 or cPRC1 in *Pcgf1*-KO IdHPCs. The situation was clearly different in ESCs, where depletion of *Pcgf1* resulted in a significant reduction of H2AK119ub1 enrichment around TSSs as well as H3K27me3 enrichment (Supplementary Fig. 3a–c). These results, therefore, prompted us to hypothesize that PCGF1-PRC1 mediates down-regulation of C1 genes in HPCs through a previously unappreciated mechanism. To examine this model, we analyzed local distribution of RING1B, H2AK119ub1, PRC2 H3K27me3, and cPRC1 in *Bcor*^ΔE9-10, *Pcgf2/4*-dKO and *Ring1a/b*-dKO IdHPCs. In *Bcor*^ΔE9-10 IdHPCs, reduction of H3K27me3 does not accompany a decrease of H2AK119ub1 marks, PRC2 or cPRC1 binding, similar to *Pcgf1*-KO (Fig. 4a, Supplementary Fig. 4a–d). This supports a role for PCGF1-PRC1 to regulate H3K27me3 deposition independent of H2AK119ub1 in IdHPCs. In contrast, H3K27me3 and H2AK119ub1 levels were unaltered despite considerable reduction of RING1B binding in *Pcgf2/4*-dKO IdHPCs, suggesting RING1A/B incorporated in cPRC1 is dispensable for H3K27me3 deposition. Importantly, in *Ring1a/b*-dKO IdHPCs, reduction of H3K27me3 was accompanied by decrease of H2AK119ub1, PRC2 and cPRC1 at C1 genes. In ESCs, the existence of

several variant PRC1 complexes, that do not possess PCGF1 but contributes to the canonical H2AK119ub1/PRC2 pathway, has already been reported[46]. We speculate that such variant PRC1 complexes, independent of PCGF1, are also active in IdHPCs and likely provide a back-up for H2AK119ub1/PRC2 mediated down-regulation of gene expression even in the absence of PCGF1. Taken together, we propose a model in which PCGF1-PRC1 regulates H3K27me3 level at C1 genes in an H2AK119ub1-independent manner. This non-canonical pathway cooperates with the H2AK119ub1/PRC2 pathway, likely downstream of other variant PRC1 complexes harboring PCGF3/5/6 but not PCGF1. Importantly, cPRC1 sub-complexes appear to be dispensable during this process.

To further consolidate this model, we asked whether the H3K27me3 marks observed in C1 genes per se were involved in transcriptional down-regulation. To this end, we cultured ERT2-Cre:*Pcgf1*^fl/fl IdHPCs in the presence of an EZH1/2 inhibitor (UNC1999) for 4 days and examined the transcriptional status and local depositions of SUZ12 and H3K27me3 at C1 and C2 genes. To this end, we determined the optimal concentration (1 μM) of UNC1999 that did not affect cell growth (Supplementary Fig. 4e). After 4 days of culture with 1 μM UNC1999, we found that local H3K27me3 levels were significantly reduced. This reduction, however, was not accompanied by loss of SUZ12 at PCGF1 target genes (Fig. 4b). The reduction in H3K27me3 levels was associated with significant up-regulation of the C1 genes (Fig. 4c). These results showed a critical role of H3K27me3 to down-regulate C1 genes. We, however, also noticed that down-regulation of H3K27me3 level and up-regulation of C1 genes expression in *Pcgf1*-KO and *Bcor*^ΔE9-10 IdHPCs were only modest in comparison with those in UNC1999-treated or *Ring1a/b*-dKO IdHPCs. This indicates that PCGF1-PRC1 and other variant PRC1 sub-complexes likely compensate each other to mediate H3K27me3 marks at C1 genes in IdHPCs.

## C1 genes down-regulated by PCGF1-PRC1 are involved in maintenance of B cell fate

We then examined whether this mode of PCGF1-dependent regulation of H3K27me3 deposition at the C1 genes revealed in IdHPCs is also active in primary HPCs. To this end, we isolated LMPP or LSK cells from BM of *Pcgf1*^fl/fl (control) and ERT2-Cre:*Pcgf1*^fl/fl (*Pcgf1*-KO) mice and cultured them for 4 days in the presence of 4-OHT and examined transcription profiles and local deposition of RING1B, SUZ12 and H3K27me3. In general, up-regulated genes in LMPPs tended to be up-regulated in IdHPCs (61%: 549 out of 893 genes) and this trend was more clearly observed in the C1 genes (73%: 54 out of 74 genes) (Supplementary Fig. 5a–d). We indeed found up-regulation of C1 genes that included myeloid- and stemness-related factors in *Pcgf1*-KO LMPPs in a manner similar to *Pcgf1*-KO IdHPCs (Fig. 5a, b, Supplementary Fig. 5a–c, e, left). In contrast, the expression of B cell-related genes was barely affected (Supplementary Fig. 5a–c, e, right). This result was confirmed by RT-qPCR for selected C1 (*Hmga2, Rxra*) and C2

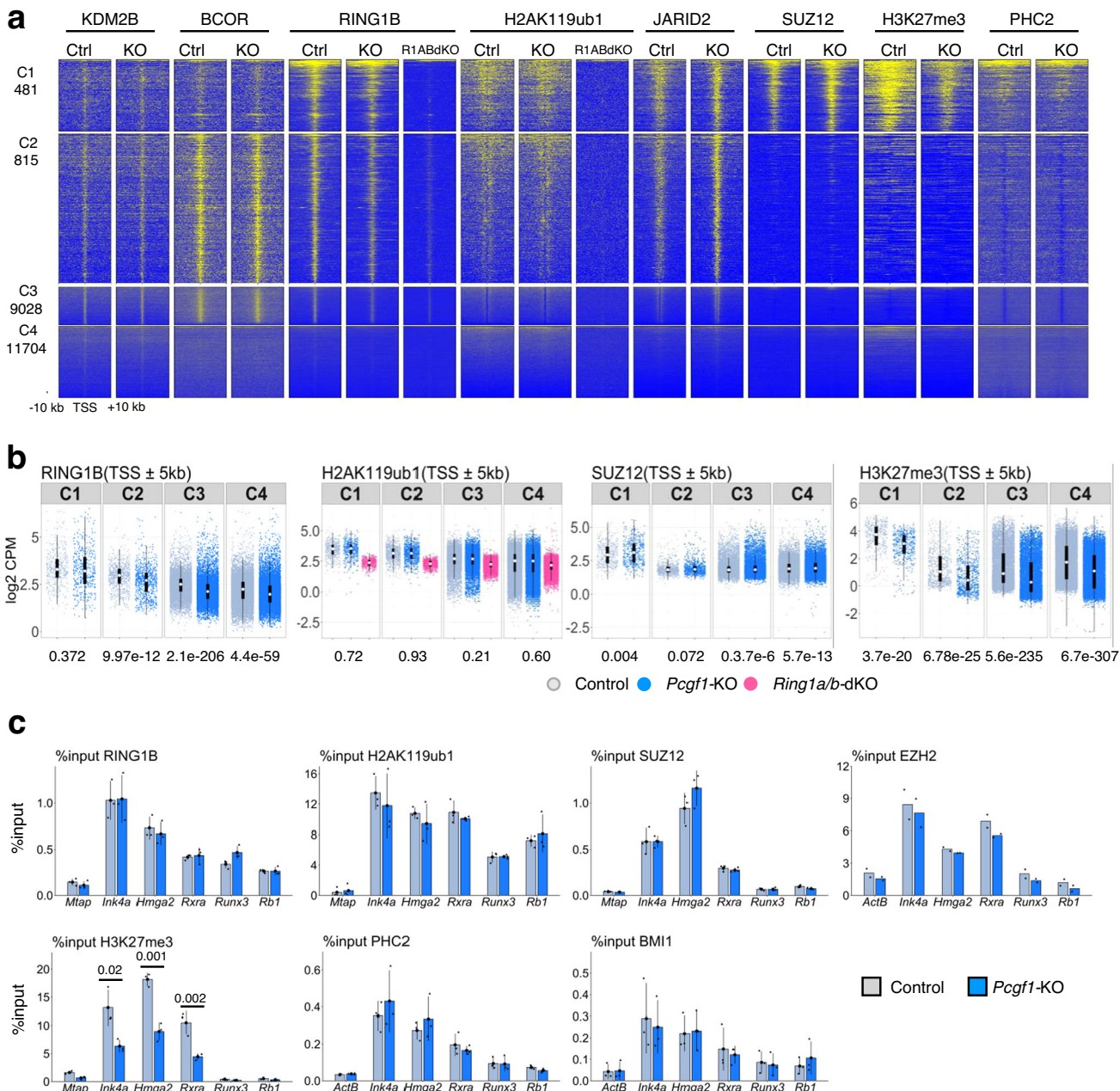

**Fig. 3 | PCGF1 is involved in stabilizing H3K27me3 at TSS regions without impact on the enrichment of H2AK119ub1 and SUZ12 in target genes of IdHPCs.** **a** Target binding of KDM2B, BCOR, RING1B, H2AK119ub1, JARID2, SUZ12, H3K27me3 and PHC2 in control (Ctrl) and *Pcgf1*-KO (KO) IdHPCs. A heatmap of ChIP-seq signals across TSS(±10 kb) of C1, C2, C3, and C4 genes in control and *Pcgf1*-KO is shown. Local levels of RING1B and H2AK119ub1 were also tested in *Ring1a/b*-dKO (R1ABdKO) IdHPCs as controls. H3K27me3 ChIP-seq were calibrated by spike-in chromatin. Representative data of biological duplicates are shown except for RING1B ChIP-seq in *Ring1a/b*-dKO IdHPCs, which was obtained from a single experiment. **b** Box plot views for ChIP-seq results across TSS ( ± 5 kb) for RING1B, H2AK119ub1, SUZ12 and H3K27me3 in each cluster in control, *Pcgf1*-KO and

(in the case of H2AK119ub1) *Ring1a/b*-dKO IdHPCs. Data in graphs represent means for two biologically independent experiments. The center circle indicates a median value and the boxes indicate 25th to 75th percentile. Each dot represents individual genes. The numbers beneath the graph are *p*-values between the control and *Pcgf1*-KO calculated with the Wilcoxon signed rank test. CPM: Counts Per Million. **c** ChIP-qPCR analyses for local binding of RING1B, H2AK119ub1, SUZ12 (PRC2), EZH2 (PRC2), H3K27me3, PHC2 (cPRC1) and BMI1 (cPRC1) at selected C1 and C2 genes in the control and *Pcgf1*-KO IdHPCs. Data represent mean ± SD of three independent experiments, except for ChIP for EZH2 which is derived from two independent analysis. The numbers on the graph are *p*-values between the control and *Pcgf1*-KO calculated with the Student's two-sided *t* test.

(*Runx3, Rb1*) genes (Fig. 5c). We tested whether up-regulation of C1 genes was accompanied by a decrease in H3K27me3 levels. ChIP-seq analysis revealed that H3K27me3 was markedly decreased at the same genes in *Pcgf1*-KO LSK cells, while RING1B occupancy did not change (Fig. 5d, Supplementary Fig. 5f, g). We further confirmed these results by visual inspection of ChIP-seq data at the *Hmga2* locus and by ChIP-qPCR for RING1B, SUZ12, H2AK119ub1, and H3K27me3 at selected C1

(*Ink4a, Hmga2, Rxra*) and C2 (*Runx3, Rb1*) genes (Fig. 5e, Supplementary Fig. 5h). Based on these findings, we conclude that PCGF1-PRC1 contributes to down-regulation of C1 genes via regulation of H3K27me3 deposition in primary HPCs, in a similar fashion to IdHPCs.

We went on to determine whether PCGF1-dependent down-regulation of C1 genes contributed to proper HPCs differentiation. Indeed, we found that various myeloid-related genes, such as *Hmga2*,

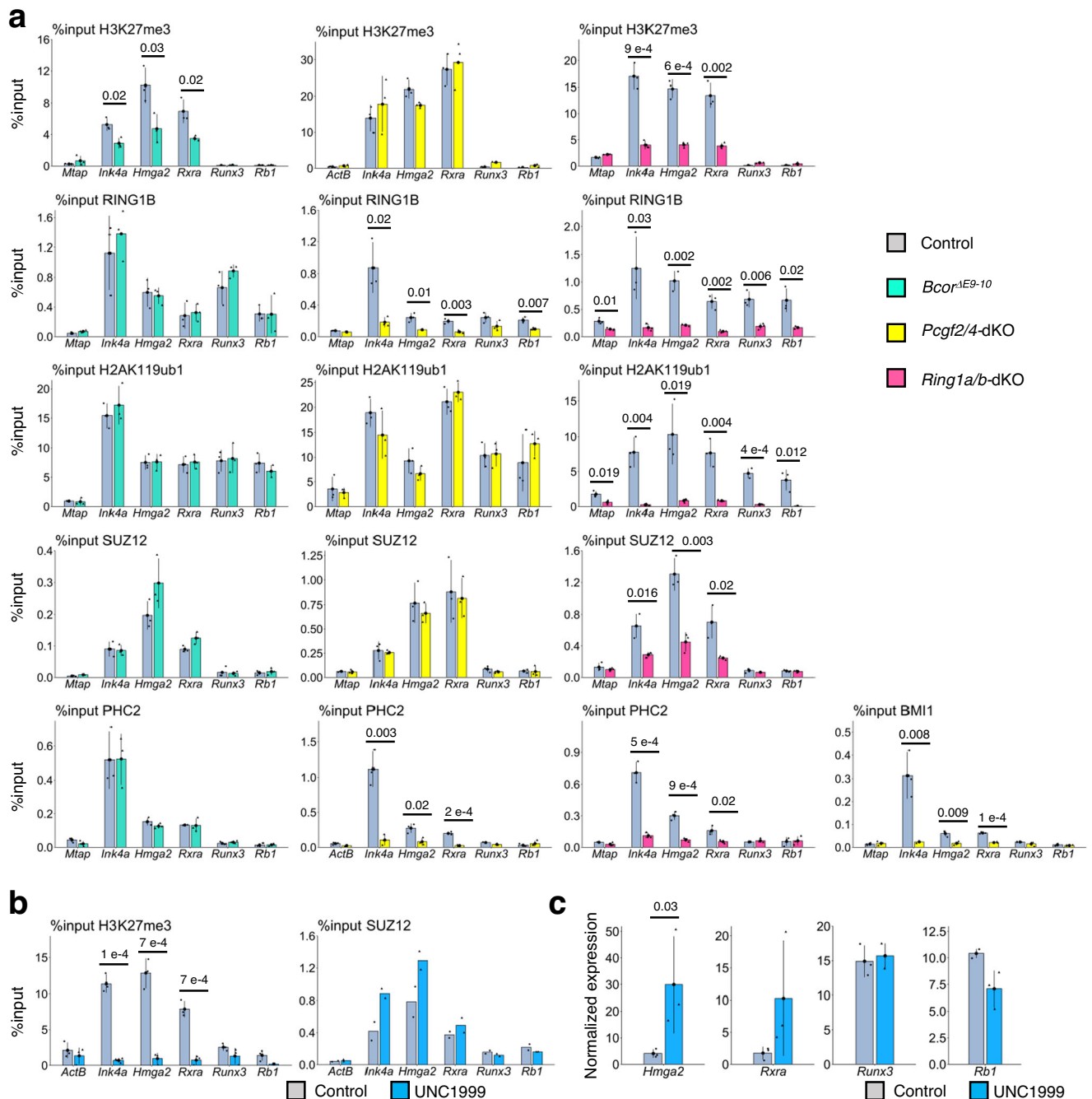

**Fig. 4 | PCGF1 maintains H3K27me3 mediated down-regulation of target genes as a component of the PCGF1-PRC1. a** Destabilization of H3K27me3 mark in $Bcor^{\Delta E9\text{-}10}$ and $Ring1a/b$-dKO IdHPCs but not in $Pcgf2/4$-dKO. ChIP-qPCR analyses for local binding of RING1B, H3K27me3, and H2AK119ub1 at selected C1 and C2 genes in the control and $Bcor^{\Delta E9\text{-}10}$, $Pcgf2/4$-dKO and $Ring1a/b$-dKO IdHPCs are shown. Data represent mean ± SD of three independent experiments. The numbers on the graph are $p$-values between the control and respective mutants calculated with the Student's two-sided $t$ test. **b, c** H3K27me3 at the C1 promoters contribute down-regulation of target genes. **b** Decrease of local H3K27me3 levels at selected C1 genes, $Ink4a$, $Hmga2$, and $Rxra$, by UNC1999. Local binding of SUZ12 to these genes was not affected. **c** Up-regulation of two selected C1 genes, $Hmga2$ and $Rxra$, upon UNC1999 treatment. The expression of selected C2 genes, $Runx3$ and $Rb1$, were not altered. Data in **b** and **c** represent mean ± SD of three independent experiments, except for ChIP for SUZ12 in **b** which is derived from two independent analysis. The numbers on the graph **b** and **c** are $p$-values between the control and UNC1999 treatment calculated with the Student's two-sided $t$ test.

were up-regulated in $Pcgf1$-KO HPCs. These genes could potentially skew the differentiation of HPCs to the myeloid lineage. We asked whether up-regulation of $Hmga2$, which facilitates myeloid lineage differentiation of HPCs[49], contributed to the biased differentiation in $Pcgf1$-KO HPCs. We isolated LSK cells from fetal liver of ERT2-Cre:$Pcgf1^{fl/fl}$ ($Pcgf1$-KO) or $Pcgf1^{fl/fl}$ (control) mice and cultured these in the presence of 4-OHT. To knockdown $Hmga2$ expression, we separately transduced control or $Pcgf1$-KO cells with two different shRNAs

targeted to $Hmga2$ and then continued culturing the cells on TSt-4 stroma to promote B cell differentiation (Fig. 5f, Supplementary Fig. 5i). We found that the decrease of B cells, and converse increase of myeloid lineage cells in $Pcgf1$-KO cells was partially restored by knocking down $Hmga2$ (Fig. 5g, h, Supplementary Fig. 5i, j). These results show that the down-regulation of $Hmga2$ by PCGF1 in HPCs is a functionally important process to restrain the myeloid cell fate and, reciprocally, to facilitate B cell development. Therefore, PCGF1-

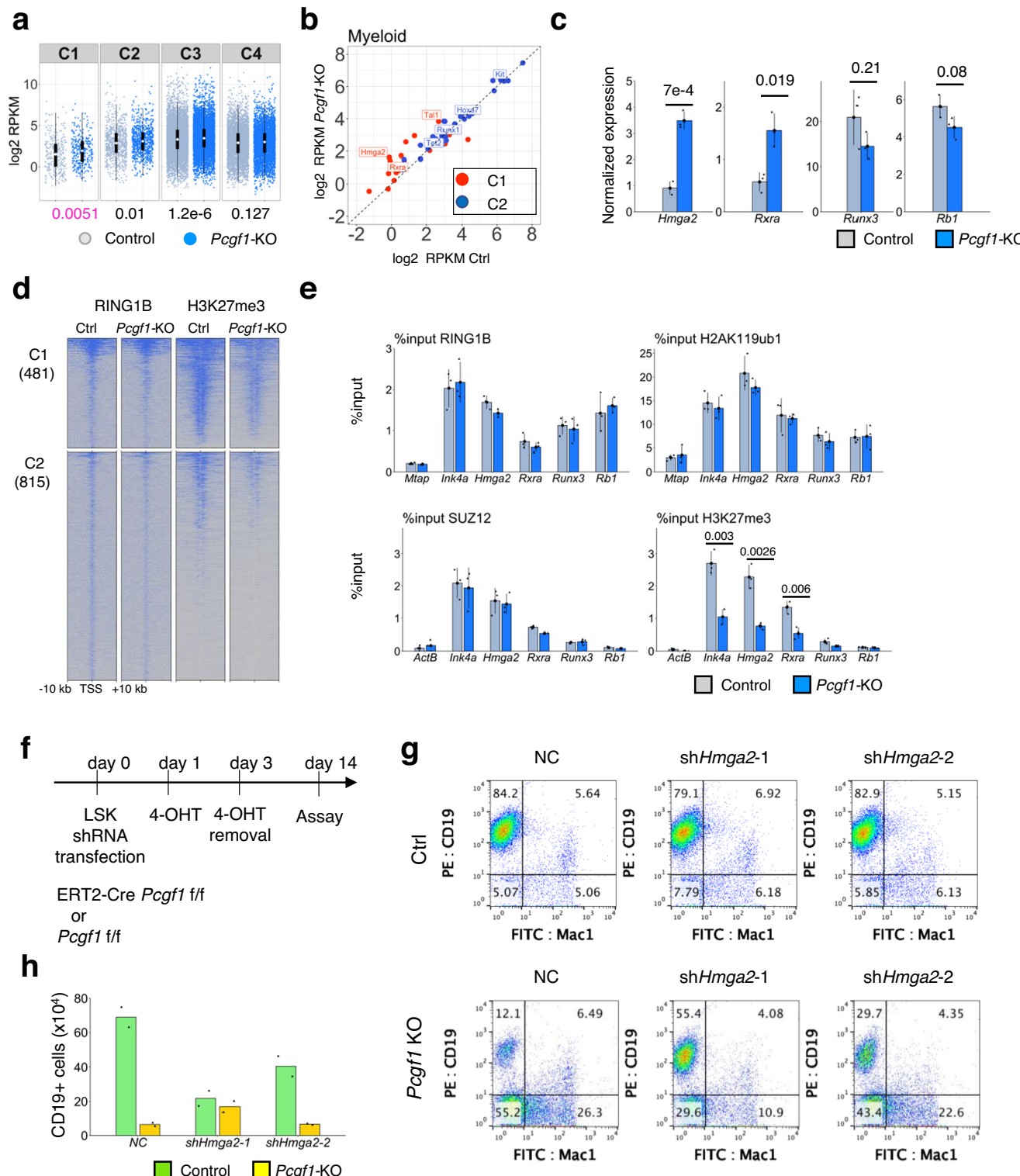

dependent down-regulation of the C1 genes via enrichment of H3K27me3 in primary HPCs plays a role to suppress premature activation of the myeloid program and to safeguard their lymphoid potential.

## PCGF1-PRC1 mediated regulation of H3K27me3 is associated with nucleosome configuration

Our observations indicated that PCGF1-PRC1 regulates H3K27me3 deposition at the C1 genes without changing PRC2 occupancy in IdHPCs and primary HPCs. We wondered whether this phenomenon

was mediated by perturbation in the PRC1 complex formation in the absence of *Pcgf1*. Indeed, immunoprecipitation (IP) of RING1B combined with IB analysis revealed the dissociation of the accessory factor RYBP from PCGF1-PRC1, indicating that a defective PRC1 complex may not properly maintain H3K27me3 deposition at C1 genes in the absence of *Pcgf1* (Supplementary Fig. 6a).

We also wondered how this *Pcgf1*-deficient PRC1 complex affects local H3K27me3 deposition at C1 genes. To test whether the absence of PCGF1 and RYBP caused defects in formation of PRC2 sub-complexes, we performed IP of SUZ12, a key component of PRC2, followed by IB

**Fig. 5 | PCGF1 contributes to facilitate H3K27me3 deposition and down-regulation of C1 genes in primary HPCs, which safeguards B lineage development. a** Box plot of gene expression in each cluster in the control and *Pcgf1*-KO LMPPs. Data in graphs represent mean for two biologically independent experiments. The center circle indicates a median value and the boxes indicate 25th to 75th percentile. Each dot indicates individual genes. The numbers below the graph are *p*-values calculated with the Wilcoxon signed rank test. **b** Gene expression profiling of "Myeloid" related genes in the control and *Pcgf1*-KO LMPPs. Data in graphs represent mean for two independent biological experiments. Red and blue dots represent C1 and C2 genes, respectively. **c** Up-regulation of C1 genes in *Pcgf1*-KO LMPPs shown by qRT-PCR analysis. Data represent mean ± SD of three independent experiments. The numbers on the graph are *p*-values between the control and *Pcgf1-KO* calculated with the Welch's two-sided *t* test. **d** A heatmap of ChIP-seq signals for RING1B and H3K27me3 across TSS (± 10 kb) of C1 and C2 in control and

*Pcgf1*-KO LSK cells. H3K27me3 ChIP-seq was calibrated by spike-in chromatin. **e** ChIP-qPCR analyses for local binding of RING1B, H2AK119ub1, SUZ12, and H3K27me3 at selected C1 and C2 genes in the control and *Pcgf1*-KO LMPPs. Data represent mean ± SD of three independent experiments. The numbers on the graph are *p*-values between the control and *Pcgf1-KO* calculated with the Welch's two-sided *t* test. **f** Schematic representation of the experimental procedure. LSK cells sorted from fetal livers of ERT2-Cre;*Pcgf1*fl/fl or *Pcgf1*fl/fl mice were infected with either sh*Hmga2-1* or sh*Hmga2-2* retrovirus and treated with 4-OHT for the next 2 days. Eleven days later, the cells were analyzed by FACS. **g** Flow cytometric profiles for CD19 and Mac1 in respective conditions are shown. Data shown in the graphs are representative of independent biological duplicates. **h** Partial restoration of total cell number of *Pcgf1*-KO CD19+ cells in each femur by sh*Hmga2-1*. In contrast, the impact of sh*Hmga2-2* was limited. Bar graphs represent mean ± SD of two independent experiments.

for AEBP2, JARID2 or PALI1 (Supplementary Fig. 6b), as association of these accessory proteins to PRC2 is reported to be context-dependent[7,8,50]. We, however, did not find any differences between control and *Pcgf1*-KO IdHPCs in binding of these proteins to SUZ12. In addition, localization of PCL2 (one of the representative accessory proteins of PRC2.1) was not affected by *Pcgf1* deletion (Supplementary Fig. 6c, left). We also examined if *Pcgf1* ablation led to enrichment of H3K27 demethylases to C1 genes. However, ChIP-seq for UTX showed no difference between control and *Pcgf1*-KO (Supplementary Fig. 6c, right).

This led us to hypothesize that PCGF1 may control H3K27me3 deposition through previously unknown mechanisms. To explore this possibility, we surveyed the PCGF1 interacting partners in IdHPCs by pull-down with an exogenous 3xFLAG-PCGF1 expressed in *Pcgf1*-KO IdHPCs, followed by mass spectrometry (IP-MS). As expected, both analyses confirmed the association of 3xFLAG-PCGF1 with RING1A/B, RYBP, BCORL1, and KDM2B (Fig. 6a, Supplementary Fig. 6d). In contrast, we could not detect PRC2 components in pull-down samples. As PRC2 complexes are normally formed and bind to C1 genes in *Pcgf1*-KO IdHPCs, we suspect that PCGF1-PRC1 affects PRC2 activity in an indirect manner (Fig. 6a, Supplementary Fig. 6b, c).

Intriguingly, however, MS analysis revealed enrichment of proteins directly involved in DNA synthesis and nucleosome formation, such as DNA helicase complexes (MCM2-7), remodeling factors, histone chaperones (RUVBL2, ANP32E and SUPT16)[51], and histones (Fig. 6a, Supplementary Fig. 6d, e). This observation was confirmed by pull-down experiments with endogenous TY1 tagged PCGF1 in IdHPCs (Supplementary Fig. 6f).

This finding indicated that PCGF1 might associate with the DNA replication machinery in IdHPCs. We therefore examined whether PCGF1 associates with the replication fork by a Proximity Ligation Assay (PLA)[20] by labeling newly synthesized DNAs with EdU, and confirmed the accumulation of PCGF1 on EdU-stained nascent DNA (Fig. 6b, Supplementary Fig. 6g). To further examine its localization on nascent DNA, we performed PCGF1 ChIP-seq on nascent DNA, by combining the Isolation of Proteins On Nascent DNA (iPOND)[52] and ChIP-seq methods (iPOND-ChIP) (Fig. 6c). This analysis revealed that PCGF1 was bound to almost all C1 and C2 genes, even in the nascent DNA, a finding indicating that PCGF1 binding to target gene promoters is not destabilized by the passage of the replication machinery. PCGF1 may therefore play a key role to mediate PcG-dependent gene repression by remaining associated with PcG target sites during DNA replication.

To reveal the role of PCGF1 in the vicinity of replication fork, we compared the association of proteins to nascent DNA in Ctrl or *Pcgf1*-KO by performing iPOND followed by mass spectrometry (iPOND-MS analysis) or an IB assay. As expected, PCGF1 and H2AK119ub1 were both detected at the replication fork (Fig. 6d, e), indicating that PCGF1 associates with nascent DNAs as a component of PCGF1-PRC1. Furthermore, we noted that PCGF1-interacting partners such as RING1B,

MCM2-7, ANP32E, SUPT16, and others also associated with nascent DNA (Fig. 6d). Importantly, binding of a group of nascent DNA-associated proteins at the fork was altered upon *Pcgf1* deletion (Fig. 6d). In particular, we found overloading of chromatin remodelers, such as BRG1 (encoded by *Smarca4*), and other activators (e.g. TAF4B) on nascent DNA in the *Pcgf1*-KO.

Next, we compared the loading of BRG1 on nascent or post-replicated mature DNA, in control or *Pcgf1*-ablated cells (Fig. 6e). Immunoblot analysis on iPOND samples revealed that overloading of BRG1 occurred specifically on nascent DNA (Fig. 6e). Furthermore, ChIP-qPCR analysis showed that local enrichment of BRG1 in the TSS of representative C1 and C2 genes in steady-state cells did not change upon *Pcgf1* deletion (Fig. 6f), supporting the hypothesis that the molecular antagonism between PCGF1 and BRG1 occurs on nascent DNA. Consistent with this notion, degradation of DNA and RNA by benzonase did not impair the interaction between PCGF1 and PCNA or BRG1, demonstrating that the association of PCGF1 and DNA replication related factors was direct, and not dependent on nucleosomes or DNA (Supplementary Fig. 6h).

We wondered whether overloading of chromatin remodelers alter the nucleosome configuration on nascent DNA. Of note, it has been shown that the enzymatic activity of PRC2 is affected by nucleosome density[38]. We therefore examined whether nucleosome density on nascent DNA was perturbed in *Pcgf1*-KO IdHPCs. We also asked if this alteration could be restored by degradation of BRG1 by treatment with the BRG1-specific PROTAC ACBI1 (MedChemExpress HY-128359)[53] (Supplementary Fig. 6i, j). To examine these questions we combined MNase-seq and iPOND (iPOND-MNase-seq)[54] techniques. We digested nucleosomes by MNase, collected mono-nucleosome derived fragments and performed deep sequencing. In particular, we focused on the −1 to +1 kb region around the TSS, where PCGF1 predominantly binds. Consistent with a previous report[54], we found a peak at the first nucleosome (+1) position after the TSS in each cluster even on nascent DNA. We could also detect nucleosome-depleted regions (NDR) around the TSS (Fig. 6g). Importantly, upon PCGF1 depletion, we observed a clear reduction of nucleosome occupancy at the +1 nucleosome in the C1, C2 and C3 genes, which was partially restored by degradation of BRG1 for 24 h (Fig. 6g). These results indicated that PCGF1 protects against destabilization of nucleosome density by BRG1 immediately after the passage of replication fork. We then asked if this alteration of nucleosome occupancy at the +1 position on nascent DNA was observed in steady-state cells. To this end, we repeated MNase-seq in IdHPCs, and found similar changes in nucleosome density at C1, C2, and C3 gene promoters (Fig. 6h). Based on these findings, we propose that PCGF1 regulates PRC2-mediated H3K27me3 deposition on nascent DNA, at least in part by regulating nucleosome density. Importantly, this function of PCGF1 involves prevention of overloading of chromatin remodelers such as BRG1 on nascent DNA. We further validated this model by performing ChIP and gene expression experiments, which revealed that reduction of H3K27me3 marks at the

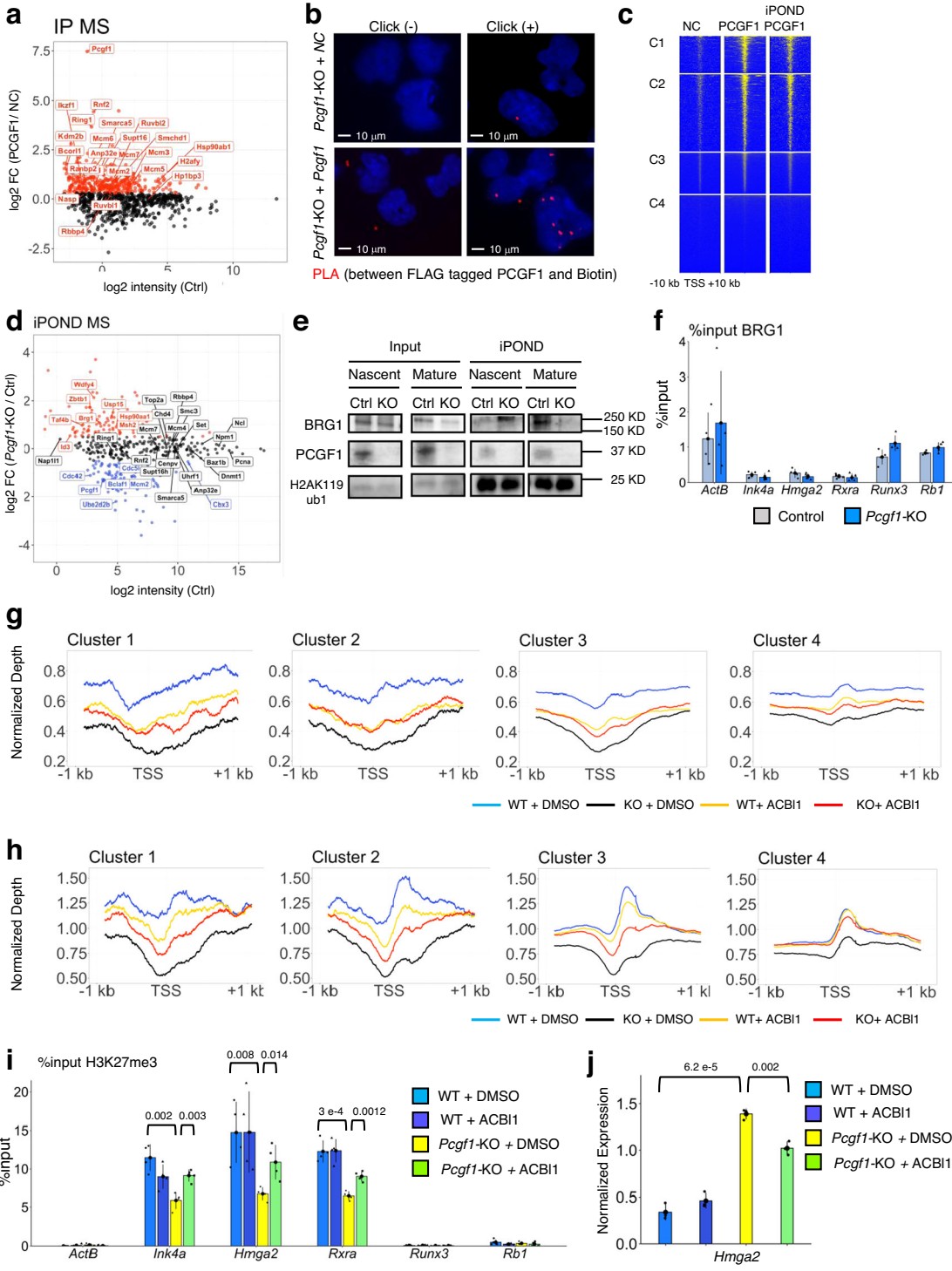

representative C1 genes (*Ink4a, Hmga2*, and *Rxra*) and de-repression of *Hmga2* upon *Pcgf1* deletion were partially restored by degradation of BRG1 (Fig. 6i, j).

Finally, we asked whether PCGF1-dependent stabilization of H3K27me3 shown in IdHPCs via regulation of nucleosome occupancy was also observed in primary HSPCs. To this end, we collected control and *Pcgf1*-KO LSKs and performed MNase-seq. We found that nucleosome density at the TSS of C1 and C2 genes was decreased (Supplementary Fig. 6k), as expected. Taken together, we propose that PCGF1-PRC1 preserves cellular identities of HSPCs through H3K27me3 mediated down-regulation of target genes via optimization of nucleosome configuration in a replication-coupled manner.

## Discussion

In this study, we reveal a previously unknown role of PCGF1 that involves antagonization of BRG1, a well-known ATP-dependent chromatin remodeler, in association with the replication fork passage. By this mechanism, PCGF1-PRC1 regulates proper nucleosome density at the replication fork to facilitate downstream PRC2 mediated repression of myeloid-lineage genes in HSPCs. Supporting this model, we demonstrate that PCGF1 associates with nascent DNA and also found that PCGF1 ablation led to the formation of a defective PCGF1-PRC1 complex in which the accessory factor RYBP is dissociated, in turn leading to overloading of BRG1 in the vicinity of replication fork. Consistent with this observation, the nucleosome density at target

**Fig. 6 | Association of PCGF1 with the replication fork limits access of chromatin remodelers. a** A scatterplot view for PCGF1 interactors in IdHPCs. The X and Y axes denote log2 signal intensities of proteins in the mock immunoprecipitation (the negative control: NC) and log2 ratios of the label-free quantification intensities of immunoprecipitated materials over the negative control (log2 FC), respectively. Orange font denotes proteins with log2 FC > 0.3. Data represent the average of two independent experiments. **b** Proximity Ligation Assay (PLA) revealed association between 3xFLAG-tagged PCGF1 and EdU-labeled nascent DNA. *Pcgf1*-KO IdHPCs with or without exogenous *Pcgf1* were used. PLA, red; DAPI, blue. Representative data from biological triplicates are shown. **c** A heatmap of ChIP-seq and iPOND-ChIP signals for endogenous TY1-PCGF1 across TSS ( ± 10 kb) in IdHPCs. Data represent the average of two independent experiments. **d** Scatterplot of iPOND-MS in control and *Pcgf1*-KO IdHPCs. The X and Y axes indicate log2 intensities of the control and log2 ratios of signal intensity of *Pcgf1*-KO over the control, respectively. Representative data from two independent biological duplicates are shown. Blue and red fonts, respectively, indicates proteins with log2 FC < −0.4 and log2 FC > 0.4.

**e** Suppression of BRG1 overloading to nascent DNA by PCGF1 in IdHPCs. iPOND materials without (Nascent) or with thymidine chase (Mature) were examined by respective immunoblotting. Representative data from biological duplicates are shown. **f** ChIP-qPCR for BRG1 in control and *Pcgf1*-KO IdHPCs. Data represent mean ± SD of three independent experiments. **g, h** Destabilization of nucleosomes on nascent DNA **g** and steady state DNA **h** around TSS in *Pcgf1*-KO IdHPCs and its partial restoration by drug-induced degradation of BRG1. **g** Results of iPOND-MNase-Seq in respective conditions. **h** Results of MNase-Seq. Data are the average of two independent experiments. **i** Partial restoration of H3K27me3 destabilization around the TSS of selected C1 genes in *Pcgf1*-KO IdHPCs by degradation of BRG1 revealed by ChIP-qPCR. Data represent mean ± SD of four independent experiments. **j** Partial restoration of *Hmga2* up-regulation in *Pcgf1*-KO IdHPCs by BRG1 depletion revealed by qRT-PCR. Data represent mean ± SD of three independent experiments. *p*-values between the control and *Pcgf1*-KO calculated with the Student's two-sided *t* test are shown.

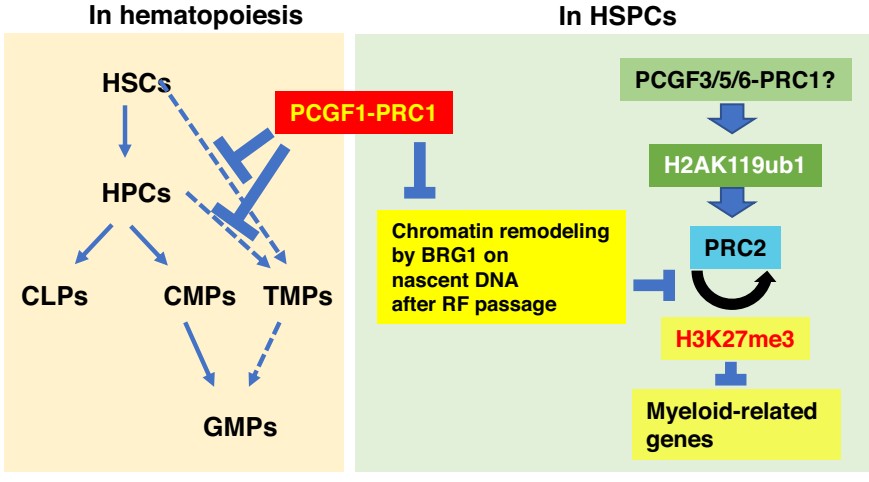

**Fig. 7 | Model depicting how PCGF1-PRC1 contributes to regulate early hematopoiesis.** Two distinct PRC1 pathways, replication-related and H2AK119ub1-dependent pathways, are cooperatively used to facilitate early hematopoiesis as shown in the right panel. PCGF1-PRC1 prevents excess accession of chromatin remodeling factors, such as BRG1, to the nascent DNA in replication-associated manner and, in turn, facilitate H3K27me3 deposition by PRC2 via maintenance of proper nucleosome density. Other variant PRC1 incorporating PCGF3/5/6 are expected to facilitate PRC2 loading via H2AK119ub1-dependent manner. PCGF1-PRC1 contributes to maintain proper identities of HSPCs by inhibiting the inappropriate emergence of TMPs as depicted in the left panel.

genes in *Pcgf1*-KO cells is reduced on nascent DNA and this reduction is partially restored by degradation of BRG1. Our results therefore clearly demonstrate that the PCGF1-PRC1 repressive complex and BRG1-mediated activation complexes compete with each other in the vicinity of replication fork, revealing a previously unknown layer of PcG-TrxG antagonism. Consistent with our observations, interaction between BRG1 and PCNA, which suggests localization of BRG1 in the vicinity of replication fork, has been demonstrated before[55]. Further supporting our model, a previous study has shown that overexpression of BRG1 in mouse LSK cells leads to a skewing towards the myeloid lineage at the expense of the lymphoid lineage[13]. We speculate that upon ablation of *Pcgf1*, myeloid-related genes are exposed to intrinsic activators, which may involve BRG1, and thereby are transcriptionally up-regulated. The activation of these genes, in turn, plays a role to drive improper differentiation of TMPs and a reciprocal exhaustion of lymphoid progenitors (Fig. 7). PCGF1-PRC1, therefore, could prevent inappropriate activation of myeloid-related genes in a DNA replication-associated manner to preserve multipotency of HSPCs. Abnormal regulation of DNA replication coupled processes could be linked to tumorigenesis, as shown by the tumor suppressive role of BCOR and the paradoxical

roles of KDM2B in prevention or promotion of malignant transformation of HSPCs[31–34,56,57]. Therefore, the antagonism between PCGF1-PRC1 and BRG1 at the replication fork could also serve to prevent malignant transformation of HSPCs.

Surprisingly, we find that PCGF1-PRC1 regulates H3K27me3 levels at C1 genes, but not local deposition of H2AK119ub1, PRC2 or cPRC1. This observation is clearly different from previous insights gained in ESCs, in which all of the above mechanisms are interlinked (Supplementary Fig. 3 a-c). What then is the molecular basis for this disparity? It is worth noting an essential difference between ESCs and HPCs; ESCs are tightly captured in a strict developmental window to undergo repetitive proliferation without changes in cellular fate, while HPCs maintain some degree of multipotency even while differentiating. Thus, the gene control mechanisms could well be different between these two cell types. In ESCs, H2AK119ub1/PRC2 axis has a predominant function to maintain pluripotency by robust silencing of development/differentiation-related genes[6,58]. In contrast, HPCs may utilize a combination of two distinct PcG pathways, the first involving a replication coupled mechanism mediated by PCGF1-PRC1 and the second involving the H2AK119ub1/PRC2-associated mechanism likely

initiated by other variant PRC1 sub-complexes incorporating PCGF3/5/6. Supporting this notion, the degree of H3K27me3 down-regulation and concomitant up-regulation of C1 gene expression in *Pcgf1*-KO or *Bcor*[ΔE9-10] IdHPCs were milder in comparison with *Ring1a/b*-dKO. The PCGF3/5/6 containing PRC1 sub-complexes, therefore, may cooperate with PCGF1-PRC1 to down-regulate C1 genes by maintaining H3K27me3 levels (Fig. 7). Combination of two distinct PRC1-related mechanisms may enable HPCs to maintain multipotency to avoid premature exhaustion of progenitor pools and at the same time to execute lymphoid-directed differentiation programs associated with cell proliferation (Fig. 7). Importantly, we also find that PCGF1-PRC1 is dispensable for recruitment of cPRC1. Although cPRC1 proteins bind to target genes by recognizing H3K27me3 in ESCs, in HSPCs H3K27me3 recognition could be therefore mediated by other readers such as SP140[59].

We show here that, 60% of PCGF1-PRC1 target genes (C2 genes) in HSPCs are occupied by H3K27ac and RNA polymerase II but not by PRC2. Consistent with this, van den Boom et al. demonstrated binding of PCGF1-PRC1 to active genes that lack H3K27me3[60]. In these genes, PCGF1-PRC1 could possess gene control mechanisms different from regulation of PRC2 activity, mediated by optimization of nucleosome configuration, which involves competition with BRG1 in the vicinity of replication fork. Such distinct classifications of PCGF1-PRC1 targets, manifested by binding of PRC2 or the lack of it, are not observed in ESCs (Supplementary Fig. 2g). Indeed, in ESCs the C2 genes also tend to be bound by PRC2. This discrepancy indicates that the functional link between PCGF1-PRC1 and PRC2 is context-dependent and not constitutive (Fig. 2a, Supplementary Fig. 2g). In such a context-dependent scenario, H3K27 acetylation observed at the C2 genes in HSPCs could be mediated by lineage-specific transcription factors (TFs). These TFs could recruit additional co-activators such as CBP and/or p300. In support of this model, we find enrichment of binding motifs for critical TFs involved in B cell differentiation, such as BHLHB2 (a motif common with E2A), ZFX, PAX2 (likely common with PAX5), MYC, and MYCN, in the C2 genes. Consistently, we also observed increased binding of MYC, E2A and PAX5 in hematopoietic lineage cells at the same group of genes (Supplementary Fig. 6l, m). This may imply that binding of such TFs facilitates the expression of the C2 genes in stage- and/or tissue-specific manner and, thereby, restrain access of PRC2 to target CGIs via active transcription[61]. In other words, down-regulation of stage/tissue-specific TFs upon differentiation may allow PRC2 recruitment, which facilitate robust down-regulation of target genes. As PCGF1-PRC1 facilitates PRC2 recruitment[26], we speculate that constitutive binding of PCGF1-PRC1 to the C2 genes regulate their timely downregulation during differentiation of HSPCs. To examine this idea, we compared the transcriptomes of LMPPs, GMPs and bone marrow macrophages obtained using public data[62] (GSE116177) and found that C2 gene expression showed a propensity to be downregulated upon myeloid-skewed differentiation (Fig. 6n).

Our findings, therefore, show that PRC1 and PRC2 are linked via previously unknown mechanisms in HSPCs to maintain proper differentiation potential. Such links, involving competition between PCGF1-PRC1 and the SWI/SNF complex may specifically occur at the replication fork.

## Methods

### Mice

C57BL/6 (B6) mice were purchased from CLEA Japan Inc. C57BL/6-Ly5.1 mice were purchased from Charles River Japan. ERT2-Cre:*Pcgf1*[fl/fl26], ERT2-Cre, ERT2-Cre:*Pcgf2*[fl/fl]/*Pcgf4*[fl/fl46], *Ty1-Pcgf1*, and ERT2-Cre:*Ring1a*[-/-]*Ring1b*[fl/fl47] mice were generated and maintained in our animal facility. The *Bcor* mutant allele was generated and provided by Vivian J. Bardwell[32]. For the analysis of hematopoietic cells derived from fetal livers, fetuses at 14 days post-coitum (dpc) were obtained by timed mating. The day that a plug was observed

was referred to as 0 dpc. All experiments were conducted according to guidelines approved by the Institutional Animal Care and Use Committee of RIKEN's Yokohama Institute. The housing conditions are as follows. Light cycle: A 14-hour light/10-hour cycle. Temperature and humidity: 18–23 °C with 40–60% humidity.

### Generation of *Ty1-Pcgf1* mice

The *Ty1-Pcgf1* allele was generated by the Alt-R CRISPR-Cas9 system (IDT) (https://sg.idtdna.com/pages/products/crispr-genome-editing/alt-r-crispr-cas9-system) with AATACAGTGTGAAAGAGAAG(AGG) as the target sequence. A DNA sequence encoding the TY1 tag was inserted into the endogenous *Pcgf1* locus by using the donor DNA indicated below. Its homologous arms correspond to the Mm9 genome sequence on chr6:83,030,587-83,030,905, with *Ty1* fusion taking place at Mm9 chr6: 83,030,748, with slight codon usage modifications at surrounding sequences: CAATGAAACAGCTATGGCTGTCCCGCTG GTTCGGCAAGGTAAGCCGGGTGCACAGTGGGCTGAGGGGCCAGCAG GTGCTGAGAGCCCACTCACTCCTTCTTTCCCTTTTCTCCTTAGCCATC TCCTTTGCTTCTCCAATACAGTGTGAAAGAgaaagaagaGGTGGAAGT GGAGGTTCAGGAGAGGTGCACACCAACCAGGACCCCCTGGACGCCG AAGTCCATACAAATCAGGATCCTCTGGATGCCGAAGTGCACACCAA TCAGGATCCCCTGGACGCTTAGGGGCCAGGCTTGCTTCCACCCCC TTCCCACCCCTCCCCAGATATTTATGTGAAATTAACTGTGGCTTTAT TTTTTGAAATAAATGCTTTTAAAAAGCACTTTTCATCTTCCTTCTTACC TGCTACACACTCAGGCTTTGGCCTGGGTCCTAATT.

### Establishment of IdHP cells

LSK cells from bone marrow were transduced with hId3 virus supernatants by spin infection, as described previously[45]. The transduced cells were maintained on TSt-4 stromal cells in the presence of 20 ng/ml IL-7 (R&D, 407-ML-025), 10 ng/ml SCF (R&D, 455-MC-010), 10 ng/ml Flt-3L (R&D, 427-FL-025).

### Sample preparation for FACS

After the harvest of the cells, the cell concentration was adjusted to under $1 \times 10^7$ cells per 100 ml with Minimum Essential Media (MEM) (ThermoFisher, 11095080) supplemented with 1% Fetal Bovine Serum (FBS) and NaHCO3. Then, antibodies were added and incubated on ice in the dark for 30 min. Cells were washed with 1 ml MEM supplemented with 1% FBS and NaHCO3. Precipitated cells were suspended in 1 ml ice-cold MEM supplemented with 1% FBS and NaHCO3 and filtered through a 37 mm nylon mesh and then run FACS (BDFACSAria with FACSDIVA 8.0.1 (BD) software were used when cells derived from BMT experiments were analyzed and when cells derived from in vitro culture were analyzed BDFACS Calliber was used.).

### RNA extraction and qRT-PCR

Total RNA was extracted by an RNeasy kit (QIAGEN, 74106) following the manufacturer's protocol. When the cell numbers were fewer than 10,000, TRIzol™ Reagent (Invitrogen, 15596026) was used. cDNA was synthesized with a VILO cDNA synthesis kit (Invitrogen, 1756050). qPCR was performed with SYBR Premix Ex Taq (TaKaRa, RR820L). Primer sequences are listed below.

| | Forward | Reverse |
|---|---|---|
| *Pcgf1* | AACTGGATCGGGTCATGCAG | TGTCTAAGCCTCGG GACTGA |
| *Ring1B* | TTGACATAGAATGGGACAGC | GTCAGCAGAAAGT CTTGTGG |
| *Hmga2* | AAGGCAGCAAAAACAAGAGC | CCGTTTTTCTCCA ATGGTCT |
| *Rxra* | CAGACATGGACACCAAACAT | CAGTGGAGAGC CGATTCC |
| *Runx3* | ACCGGCAGAAGATAGAAGAC | CTCGTGGTGCT GAGAGAG |

| | | |
|---|---|---|
| *Rb1* | GAACATCGAATCATGGAATCCCT | AGAGGACAAGCAG ATTCAAGGTGAT |
| *hARP* | CGACCTGGAAGTCCAACTAC | ATCTGCTGCA TCTGCTTG |

### RNA-seq

Poly(A) mRNA was isolated from total RNA by a NEBNext poly(A) mRNA magnetic isolation module (New England Biolabs, E7490) and then libraries were prepared by a NEBNext Ultra RNA library preparation kit for Illumina(New England Biolabs, E7530). The average size of each library was measured by an Agilent 4200 TapeStation (Agilent Technologies). The quantity of each library was measured by a KAPA library quantification kit (KAPA Biosystems, KK4828). Sequences were read by a HiSeq 1500 system (Illumina).

### RNA-seq data analysis

RNA-seq reads were aligned to the mouse reference genome (mm9) using HISAT2 (v2.1.0; http://daehwankimlab.github.io/hisat2/). Transcript levels (counts) were summarized per gene using Rsubread[63]. Count per million (CPM) and Reads Per Kilobase of exon per Million (RPKM) were calculated using edgeR[64]. Low abundance genes (CPM < 1 across all samples) were filtered out. The significance of differences in CPM was calculated based on a negative binomial model using exactTest command of edgeR and genes with FDR < 0.05 were regarded as differentially expressed genes.

### Preparation of scRNA-seq library

FACS sorted LSK cells (control and *Pcgf1*-KO) were subjected to preparation of individually barcoded single-cell RNA-Seq libraries using the Chromium instrument and the Single Cell 3′ Reagent kit following the manufacturer's instructions (10× Genomics). Nucleotide sequencing was performed by Next-seq500 (Illumina).

### Analysis of scRNA-seq data

Raw sequencing data for control and *Pcgf1-KO* LSK cells in this study were converted to fastq format using cellranger mkfastq (10× Genomics, v3.0.0). Nestrowa's data (GSE81692) were obtained from NCBI GEO. These fastq files were aligned to the mm9 reference genome and quantified using cellranger count (10× Genomics, v.3.0.0). Count matrixes were incorporated into Seurat objects and proceeded to further analysis using Seurat 3.0[65]. scRNA-seq quality control: As we wished to filter out cells whose transcripts were lowly captured and also to avoid multiplet representation, we filtered cells with UMIs lower than 200 and higher than 10000 in both control and *Pcgf1*-KO. Integrative analysis: We first log normalized the transcript counts of each sample by using NormalizeData function (normalization.method = "LogNormalize", scale.factor = 10000). Next, we identified the top 3,000 variable genes in each sample. Then, anchors were identified by FindIntegrationAnchors (dims = 1:30). These anchors were used to integrate datasets for control, *Pcgf1-KO* LSK cells and Netrowa's data together using IntegrateData function (dims = 1:30). We performed integrated analysis on all cells with these integrated datasets. First, we applied liner transformation by ScaleData function. Next, we performed PCA on the scaled data by RunPCA(npca=30) command. The same PCs obtained from this PCA were used as input to the non-linear dimensional reduction UMAP (dims=1:20) and plotted in ggplot2 using R. We then clustered cells by finding the nearest neighbors between their cells using FindNeighbors (dims=1:20) and FindClusters (resolution=0.2) functions.

### Trajectory analysis

We used Monocle (v.2.4.0) to infer the trajectory tree of all groups of cells. A CDS object was created from the above mentioned integrated datasets. The information of clusters and coordination of each cells in UMAP was also incorporated into this object. Differentially expressed genes between clusters were calculated based on a negative binomial model and decided as genes that define a cell's progress. Then data dimensionality was reduced and cells were ordered by the orderCells function. The trajectory was visualized by the plot cell_trajectory command.

### Chromatin immunoprecipitation (ChIP) assays and ChIP sequencing (ChIP-seq)

ChIP for H2AK119ub was performed as follows. One million cells were cross-linked by adding formaldehyde to 1% and incubated for 10 min at RT, quenched using 1 M glycine and washed three times in PBS. Collected cells were lysed in 1 ml PBS + 0.5% Triton X-100 for 3 min on ice. Nuclei were pelleted and washed with Micrococcal Nuclease (MNase) digestion buffer (10 mM Tris-HCl, pH 7.4, 15 mM NaCl, 60 mM KCl, 0.5 mM spermidine), spun down and resuspended in the same buffer. $Ca^{2+}$ concentration was adjusted to 1 mM with 1 M $CaCl_2$ and MNase (TaKaRa, 2910 A) was added to 60 U/ml. The reaction mixture was incubated at 37 °C for 10 min and the reaction was stopped by adding EDTA up to 2 M. After the addition of an equal volume of 150 mM NaCl-RIPA buffer (50 mM Tris-HCl pH 8.0, 1 mM EDTA, 150 mM NaCl, 0.1% SDS, 1% Triton X-100, 0.1% NaDOC, and protease inhibitor cocktail), the chromatin solution was centrifuged and the soluble chromatin fraction was immunoprecipitated with antibody-coupled Dynabeads M-280 Sheep anti-Rabbit IgG (Veritas DB11203) at 4 °C for overnight. Immunoprecipitates were washed with high salt RIPA buffer (500 mM NaCl) four times and TE buffer twice. Bound chromatin and input DNA were suspended in elution buffer (1% SDS, 0.1 M NaHCO3, 10 mM DTT, and RNaseA) and incubated at 25 °C for 15 min and then the supernatant was collected using a magnetic rack. After the addition of NaCl, the solutions were incubated at 65 °C overnight and then treated with 0.1 mg/ml proteinase K at 55 °C for 1 h to reverse crosslinking. Eluted DNA was purified with a MinElutePCR Purification Kit (QIAGEN, 28004). Immunoprecipitated and input DNA were quantified by qPCR with the primers listed below. ChIP for FLAG, RING1B, SUZ12, H3K27me3 and others were performed as described previously[27] with small modifications. Cells were crosslinked with 1% formaldehyde at RT for 10 min. Fixed cells were sonicated using a Sonics Vibracell VCX 130 processor with a 3 mm stepped microtip in a volume of 150 mM NaCl-RIPA buffer (50 mM Tris-HCl pH 8.0, 1 mM EDTA, 150 mM NaCl, 0.1% SDS, 1% Triton X-100, 0.1% NaDOC, and protease inhibitor cocktail) for 20 s × 6 pulses at 30% amplitude. Cell extracts were subjected to immunoprecipitation with antibody-conjugated Pierce protein A/G magnetic beads (ThermoFisher, 88802). Immunoprecipitates were washed with 150mM-NaCl RIPA buffer and high salt RIPA buffer (250 mM NaCl) twice and TE buffer (10 mM Tris-HCl pH 8.0 and 1 mM EDTA). Bound chromatin was eluted as described above.

| | Forward | Reverse |
|---|---|---|
| *Mtap* | TTGCCATACTGCTTGCTGAC | AACACCCAGCCTGATGCTAC |
| *Ink4a* | GATGGAGCCCGGACTACAGAAG | CTGTTTCAACGCCCAGCTCTC |
| *Hmga2* | CATCAGCCTCCTACGGGAAG | TGCGAGTCCGAAGCTCTTAG |
| *Rxra* | CTTCACCGGCCTCAGTTTCC | CAAGGCTCCCTGCAGAAGAG |
| *Runx3* | TAGTGGCATGGAAACCGGAG | GCGGTCCTCATCCCAGTTAC |
| *Rb1* | TCCTCACCCGACTCCCGTTA | GCGGAAGTGACGTTTTCCC |

In ChIP-seq, eluted DNA samples were sheared using a Covaris S220 (Covaris inc.) at 300 bp shearing and then the library was prepared by a

NEBNext Ultra II DNA library preparation kit for Illumina (New England Biolabs, E7465). The average size of each library was measured by an Agilent 4200 TapeStation (Agilent Technologies). The quantity of each library was measured by a KAPA library quantification kit (KAPA, Biosystems KK4828). Sequences were read by a HiSeq 1500 or NextSeq500 system (Illumina). In ChIP-seq for H3K27me3, 5% of spike-in of S2 Drosophila chromatin (Activ Motif, 53083) was added for calibration after the completion of the fragmentation of DNAs.

## ChIP-seq data analysis

For ChIP-seq without the spike-in genome, reads were aligned to mm9 using bowtie2[66]. ChIP-seq experiments which contained a spike-in genome were aligned against concatenated genome of mm9 and dm3 using bowtie2 and resulting SAM files were then split such that reads aligning to *mus musculus* and *drosophila melanogaster* were placed in separate files. The SAM files were converted to the BAM format using Samtools[67]. The PCR duplicates were removed by Picardtools (http://broadinstitute.Github.io/picard). The BAM files were converted to bigwig files by deepTools2[68] for visualization. For comparison across ChIP-seq samples without a spike-in genome, the bigwig files were normalized to reads per kilobase million mapped sequence reads (RPKM). For ChIP-seq with a spike-in genome, RPKM were further normalized according to normalization factors calculated as 1 over the number of reads (per million) mapping to *drosophilia* as previously reported[69]. Heatmaps were generated by deepTools2[68]. Genome browser tracks were produced by pyGenomeTracks[70]. Regions of H3K27me3 enrichment were identified using the dpeak function of DANPOS2(-q 40 -kw 750 -kd 1500, height_logP > 120)[71]. PCGF1 and RING1B peaks were generated using MACS2 broad mode[72]. Peaks were annotated by ChIPseeker[73]. Read counts within intervals were calculated by QuasR[74]. For ChIP-seq without a spike-in genome, count per million (CPM) were calculated to conduct comparisons between samples and for ChIP-seq with spike-in DNA, they were further normalized based on normalizing factors as described before. Published ChIP-seq data for KLF4, MYC, E2A and PAX5 were obtained from NCBI GEO (accession numbers GSM1324615, GSM546535 and GSM2863171 respectively). Gene ontology (GO) analyses were performed using clusterProfiler[75]. Universal genes were used as background. Selected GO terms were considered significant with FDR < 0.05.

## Statistical analysis

All statistical analyses and visualization of box-plots and scatterplots were performed using R (version 3.2.2), except for single cell RNA-seq analysis by Seurat and Monocle which required newer version of R and performed by R version 4.0.0.

## Motif analysis

The coordinates of the promoters in Cluster 1 and 2 genes were obtained from FANTOM6 databases (http://fantom.gsc.riken.jp/5/datafiles/latest/extra/CAGE_peaks/mm9.cage_peak_phase1and2combined_coord.bed.gz) and −300 to +100 regions around the representative position were extracted. Genomic sequences were obtained using the "BSgenome" package in Bioconductor (http://www.bioconductor.org/packages/2.5/bioc/html/BSgenome.html). The sequences for motifs were scanned using the "Biostrings" package (https://bioconductor.org/packages/release/bioc/html/Biostrings.html). The comparison with background (universal genes) was performed by counting numbers of motifs in the cluster and the background and doing a one-tailed Fisher's exact test. *p*-values were corrected for the multiple testing by the Benjamini-Hochberg algorithm.

## Immunoblotting

The IdHP cells ($1 \times 10^6$) were suspended with SDS sample buffer (0.068 M Tris HCl pH 6.8, 2% SDS, 5% Glycerol, 0.005% BPB and 0.1 M DTT) and sonicated 15 s at 20% amplitude using Sonics Vibracell VCX

130 processor. Cell lysates were subjected to SDS-PAGE using Mini-Protean TGX precast gels (Bio-Rad, 4561086) and then transferred to a PVDF membrane by the Transblot Turbo transfer system (Bio-Rad). Membranes were blocked by a PVDF blocking reagent for Can Get Signal (TOYOBO, NYPBR) for 4 °C overnight and then incubated with primary antibodies diluted in Can Get Signal solution 1 (TOYOBO, NKB 201) for overnight at 4 °C. The membrane was washed three times with TBS-T and then incubated with secondary antibody diluted in Can Get Signal Solution 2 (TOYOBO, NKB 301) for 1 h at room temperature. After washing four times by TBS-T, immunoreactive proteins were detected by Western Lightning Plus ECL reagent (Perkin Elmer).

## Immunoprecipitation and Immunoblot analysis

Whole cells were lysed in 0.1% NP-40 lysis buffer (250 mM NaCl) and sonicated for 10 s × 3 pulses at 30% amplitude using a Sonics Vibracell VCX 130 processor. After centrifugation, the supernatants were immunoprecipitated with the indicated antibody-coupled Dynabeads M-280 Sheep anti-Rabbit IgG (Veritas DB11203) or Dynabeads M-280 Sheep anti-Mouse IgG (Veritas, DB11201) with or without 250 Units Benzonase (Sigma-Aldrich, E1014-5KU) per mg protein. When the experiment was involved in Benzonase the buffer was supplemented with $MgCl_2$ up to 1.5 mM. Immunoprecipitates were washed by 0.1% NP-40 lysis buffer (350 mM NaCl) for four times, then proteins were separated by SDS-PAGE and analyzed by immunoblotting.

## shRNA

LSK cells sorted from fetal livers were transduced with the indicated shRNA retrovirus by spin infection and the cells were cultured on TSt4 stromal cells. At two days after the infection, cells were collected and cultured on TSt4 cells to induce B cell differentiation or sorted for hCD25+ cells by magnetic-activated cell sorting (MACS) (Miltenyi Biotec, 131-090-312) to extract RNAs. The shRNA sequences were as follows.

ShHmga2−1: TGCTGTTGACAGTGAGCGCAGGACTATATTAATCACTTTGTAGTGAAGCCACAGATGTACAAAGTGATTAATATAGTCCTTTGCCTACTGCCTCGGA

ShHmga2−2: TGCTGTTGACAGTGAGCGACTGCTAGATTGTTACATTAATTAGTGAAGCCACAGATGTAATTAATGTAACAATCTAGCAGGTGCCTACTGCCTCGGA

## Antibodies

The following antibodies were used in indicated dilution rates for FACS analysis:

| |
|---|
| APC Rat anti Mouse CD45.1 (30-F11) (BD 559864) 1 : 100 |
| PE-Cy7 Rat anti Mouse CD117 (2B8) (BD 558163) 1 : 100 |
| APC Rat anti Mouse CD117 (2B8) (BD 561074) 1 : 100 |
| PE Rat anti Mouse Ly6A/E (D7) (BD 562059) 1 : 100 |
| APC-Cy7 Rat anti Mouse Ly6A/E (D7) (BD 560654) 1 : 100 |
| PE Rat anti Mouse CD135 (A2F10.1) (BD 553842) 1 : 100 |
| CD34 Monoclonal Antibody (RAM34), FITC, eBioscience™ (Invitrogen 11-0341-81) 1 : 50 |
| APC Rat anti Mouse CD127 (SB/199) (BD 564175) 1 : 100 |
| APC anti-mouse CD93 (AA4.1, early B lineage) Antibody (C1qRp) (Biolegend 136509) 1 : 100 |
| PE Rat anti Mouse CD19 (1D3) (BD 553786) 1 : 100 |
| APC/Cy7 Rat anti Mouse B220 (RA3-6B2) (BD 552772) 1 : 100 |
| PE/Cy7 Rat anti Mouse IgM (R6-60.2) (BD 552867) 1 : 100 |
| BV421 Rat anti Mouse CD43 (S7) (BD 752957) 1 : 100 |
| PE Rat anti Mouse IgD (11-26 C) (BD 558597) 1 : 100 |
| APC/Cy7 Rat anti Mouse CD8 (53-6.7) (BD 561967) 1 : 100 |
| BV421 Rat anti Mouse CD4 (H129.19) (BD 740024) 1 : 100 |

| | |
|---|---|
| BV510 Hamster anti Mouse CD3ε (145-2C11) (BD 563024) 1 : 100 | |
| FITC Rat anti Mouse CD25 (7D4) (BD 553071) 1 : 100 | |
| PE Rat anti Mouse CD44 (IM7) (BD 553134) 1 : 100 | |
| PerCP-Cy5.5 Mouse anti Mouse NK1.1 (PK136) (BD 561111) 1 : 100 | |
| PerCP-Cy5.5 Mouse Lineage Antibody Cocktail, with Isotype Control (BD 561317) 1 : 100 | |
| PE-Cy7 Rat Anti-Mouse TER-119/Erythroid Cells (TER-119) (BD 557853) 1 : 100 | |
| FITC Rat anti Mouse Mac-1 (M1/70) (BD 557396) 1 : 100 | |
| V450 Rat anti Mouse Mac-1 (M1/70) (BD 560456) 1 : 100 | |
| PE Mouse anti Human CD25 (M-A251) (BD 55432) 1 : 100 | |

Antibodies used for IB, IP and ChIP are listed below.

| | |
|---|---|
| anti m2_FLAG(Sigma-Aldrich, F3165) | ChIP (1 : 50), IP (1 : 100), IB (1 : 1000) |
| anti Ubiquityl-Histone H2A (Lys119)(D27C4) (Cell Signaling, #8240) | ChIP (1 : 100), IB (1 : 1000) |
| anti SUZ12(D39F6) (Cell Signaling, #3737) | ChIP (1 : 50), IP (1 : 100), IB (1 : 1000) |
| anti H3K27me3 (Merck Millipore, 07-449) | ChIP (1 : 100), IB (1 : 1000) |
| anti-acetyl-Histone H3(Lys27) (Merk Millipore, 07-360) | ChIP (1 : 100), IB (1 : 1000) |
| anti-RNA polymerase II CTD repeat YSPTSPS [8WG16] - ChIP Grade (abcam, ab817) | ChIP (1 : 50) |
| anti Bmi1 (D20B7) XP Rabitt mAb (Cell Signaling, #6964) | ChIP (1 : 50) |
| anti PCGF1(E-8)(Santa Cruz Biotechnology, sc-515371) | WB (1 : 250) |
| anti EZH2(D2C9) (Cell Signaling, #5246) | ChIP (1 : 50), IB (1 : 1000) |
| anti AEBP2(D7C6x) (Cell Signaling, #14129) | WB (1 : 1000) |
| anti JARID2 (D6M9X)(Cell Signaling, #13594) | ChIP (1 : 50), IB (1 : 1000) |
| anti DEDAF Merck Millipore, AB3637) | WB (1 : 1000) |
| anti SKP1(D3J4N) (Cell Signaling, #12248) | WB (1 : 1000) |
| anti MCM7(141.2)(Santa Cruz Biotechnology, sc-9966) | WB (1 : 1000) |
| anti PCNA(PC10) (Santa Cruz Biotechnology, sc-56) | WB (1 : 1000) |
| anti RUVBL2 (Bethyl Laboratories, A302-536A) | WB (1 : 1000) |
| anti BCOR (Proteintech, 12107-1-AP) | ChIP (1 : 20), IB (1 : 1000) |
| anti TY1 (Rockland Immunochemicals Inc, 200-301-W45) | ChIP (1 : 50), IP(1 : 100) |
| anti SMARCA4/BRG1 (Proteintech, 21634-1-AP) | ChIP (1 : 50), IB (1 : 1000) |
| Anti UTX(Merck Millipore, ABE1865) | ChIP (1 : 50) |
| anti RING1B (in house) | ChIP (1 : 20), IB (1 : 250) |
| anti PHC2 (in house) | ChIP (1 : 20) |
| anti EED (in house) | IB (1 : 1000) |
| anti KDM2B (in house) | ChIP (1 : 20) |
| Anti-Mouse IgG, HRP-Linked Whole Ab Sheep(Cytiva, NA931) | IB (1 : 5000) |
| Anti-Rabbit IgG, HRP-Linked Whole Ab Donkey(Cytiva, NA934) | IB (1 : 5000) |

## Sample preparation for LC-MS/MS measurement

The sample preparation for LC-MS/MS measurement was followed as reported previously[76]. The sample was treated with 10 mM dithiothreitol at 50 °C for 30 min and then subjected to alkylation with 30 mM iodoacetamide in the dark at room temperature. The mixture was diluted 4-fold with 50 mM ammonium bicarbonate and digested using 800 ng Lys-C and 400 ng trypsin overnight at 37 °C. An equal volume of ethyl acetate was added to the digested samples, and the mixture was acidified with 0.5% trifluoroacetic acid (TFA) (final concentration) according to the PTS protocols[77,78]. The mixture was shaken for 5 min and centrifuged at 15,000 × g for 5 min for phase separation; then, the aqueous phase was retrieved. The volume of the digested sample recovered was reduced to half or less of the original volume by a centrifugal evaporator for the complete removal of ethyl acetate, and then the mixture was desalted using C18-Stage Tips[79]. The peptides trapped in the C18-Stage Tips were eluted with 40uL of 50% acetonitrile (ACN) and 0.1% TFA, followed by drying using a centrifugal evaporator. The dried peptides were redissolved in 20 μL of 3% ACN and 0.1% formic acid, 2 μL of the redissolved sample was analyzed by LC-MS/MS.

## LC-MS/MS and data analysis

A preliminary DDA (data dependent acquisition) set was performed for SWATH protein quantification as below[76]. Peptides (approximately 100 ng) were directly injected onto a 100 μm × 15 cm PicoFrit emitter (New Objective) packed in-house with 120 A porous C18 particles (ReproSil-Pur C18-AQ 1.9 μm; Dr. Maish GmbH) and then separated using 240-min ACN gradient (3 to 40%, flow rate 300 nl/min) using an Eksigent ekspert nanoLC 400 HPLC system (Sciex). Peptides eluting from the column were analyzed on a TripleTOF 5600+ mass spectrometer. MS1 spectra were collected in the range of 400–1200 m/z for 250 ms. The top 25 precursor ions with charge states of 2+ to 5+ that exceeded 150 counts/s were selected for fragmentation with rolling collision energy, and MS2 spectra were collected for 100 ms. A spray voltage of 2100 V was applied.

All MS/MS files were searched against the UniProtKB/Swiss-Prot mouse database (Proteome ID: UP000000589, downloaded October 30, 2019, 17069 proteins entries), combined with the standard MaxQuant contaminants database (http://www.coxdocs.org/doku.php?id=maxquant:start_downloads.htm), using ProteinPilot software v. 4.5 with Paragon algorithm for protein identification[76]. The search parameters were as follows: cysteine alkylation of iodoacetamide, trypsin digestion, and TripleTOF 5600. For a protein confidence threshold, we used the ProteinPilot unused score of 1.3 with at least one peptide with 95% confidence. Moreover, proteins and peptides identified as contaminating proteins were excluded. Global false discovery rate for both peptides and proteins was lower than 1% in this study.

SWATH DIA (data independent acquisitions) were performed by using the same gradient profile used for DDA experiments as describe above. Precursor ion selection was done in the 400–1200 m/z range, with a variable window width strategy (7 to 75 Da). Collision energy for each individual SWATH experiment was set at 45 eV, and 80 consecutive SWATH experiments (100–1800 m/z) were performed, each lasting 36 ms. DIA raw data were analyzed by using the SWATH processing embedded in PeakView software (SCIEX). Peptides from PIG Trypsin and Protease I precursor Lysyl endopeptidase were used for retention time recalibration between runs. The following criteria were used for DIA quantification: Peptide Confidence Threshold 99%, 30 ppm maximum mass tolerance, 6 min maximum RT tolerance. Multivariate data analysis was performed by using Markerview software (SCIEX).

## Proximity ligation assay (PLA)

PLA was performed as described[20] with minor modifications. *Pcgf1*-KO IdHPCs reconstituted with exogenous 3xFLAG-tagged PCGF1 or empty vector were labeled with 1 μM EdU for 20 min and 20,000 cells per each condition were harvested and cytospun (750 rpm for 5 min) on polylysine-coated cover glasses, fixed at room temperature with 4% paraformaldehyde in PBS for 15 min and permeabilized in 0.25% Triton. Then half of each slide was subjected to a Click reaction as described in the iPOND section. The PLA reactions (Olink) between the anti-biotin antibody and antibodies to FLAG were performed according to the manufacturer's instructions. Following PLA, cells were immunostained

with anti-biotin Alexa Flour 488 antibody (ThermoFisher, #53-9895-82).

## iPOND and mass spectrometry (iPOND-MS) analysis

iPOND was performed as described[52] with minor modifications. Cells were labeled with 1 μM 5-ethynyl-2′-deoxyuridine (EdU) (Thermo Fisher, A10044), for 20 minutes. After labeling, cells were rinsed twice with PBS and cross-linked in 1% formaldehyde/PBS for 10 min at room temperature, quenched using 1 M glycine, and washed three times in PBS. Collected cell pellets were frozen at −80 °C. 0.25% Triton-X/PBS was used to permeabilize cells. Biotin Azide (PEG4 carboxamide-6-Azidohexanyl Biotin) (Thermo Fisher, B10184) was used to conjugate with EdU via the click reaction. After lysis and sonication, DNA-protein complexes were captured by streptavidin-coupled magnetic beads (Pierce, 88816). Beads were washed four times in RIPA buffer. Captured proteins were eluted and cross-links were reversed in sample buffer (500 mM Tris-HCl pH 8.0; 24 mM sodium deoxycholate and 24 mM sodium laurylsulphate) by incubating for 20 minutes at 95 °C with shaking.

## iPOND-ChIP-seq analysis

ChIPed Chromatin was treated with Proteinase K overnight at 65 °C and purified using MinElute PCR Purification Kit (QIAGEN, 28004). Then ChIPed DNA was conjugated with biotin in 300 μl of click reaction supplied with 0.3 mM Biotin Azide (Thermo Fisher, B10184), 0.2 mM CuSO$_4$, 1 mM THPTA (Sigma-Aldrich, 762342) and 20 mM Sodium ascorbate (Sigma-Aldrich, A7631) for 30 minutes at room temperature. After ethanol precipitation the biotinylated DNA was End Repaired using NEBNext Ultra II DNA library Prep Kit for Illumina (New England Biolabs, E7465) and ligated with an adaptor from IDT according to the manufacturer's protocol. Then DNA was dual size-selected (bead ratio: 0.55× -1.8×) with SPRI beads (Beckman coulter, B23318) as described by manufacturer's instructions and eluted into 50 μl of EB. Then Edu-biotin conjugated DNA was captured with 200 μg of Dynabeads MyOne streptavidin T1 (Thermo Fisher, DB65601) for 1 h with rotation and washed sequentially with 500 μl of the following buffers at room temperature: 3 times with 1× B&W buffer (5 mM Tris-HCl pH 7.5, 1 M NaCl, 0.5 mM EDTA and 0.05% Tween20) and twice of 2× B&W buffer. ChIP-enriched nascent dsDNA fragments were amplified using Q5 Master Mix (NEBNext Ultra II DNA library Prep Kit for Illumina, NEB) kept on beads as follows: 98 ˚C 30 s, 5 cycles of (98 °C, 10 s; 65 °C 75 s), vortexed quickly and 6 cycles of (98 °C, 10 s; 65 °C 75 s), and last extension 65 °C, 5 min. Libraries were cleaned up using 1.5× (bead ratio) SPRI beads (Beckman coulter, B23318) according to the manufacturer's instructions and run on High Sensitivity DNA Screen Tape (Agilent) for quality control.

## MNase-seq analysis

MNase-seq analysis was performed according to Zhao et al.[80] with minor modifications. One million cells were cross-linked by adding formaldehyde to 1% and incubated for 10 min at RT, quenched using 1 M glycine, and washed three times in PBS. Collected cells were lysed in 1 ml PBS + 0.5% Triton X-100 for 3 minutes on ice. Nuclei were pelleted and washed by MNase digestion buffer (10 mM Tris-HCl, pH 7.4, 15 mM NaCl, 60 mM KCl, 0.5 mM spermidine), spun down and resuspended in the same buffer. Ca$^{2+}$ concentration was adjusted to 1 mM with 1 M CaCl$_2$ and MNase was added to 20 U/ml. The reaction mixture was incubated at 37 °C for 10 min and the reaction was stopped by adding 150 μL of stop buffer (20 mM EDTA, 20 mM EGTA, 0.4% SDS, 0.5 mg/ml proteinase K) and then incubated overnight at 65 °C. DNA was purified using a MinElute PCR purification Kit (QIAGEN, 28006) and loaded to 1.5% Agarose Gel. Mononucleosomal bands were excised and purified by a gel extraction kit (QIAGEN, 28705). Purified DNA was subjected to library preparation by a NEBNext Ultra II DNA library preparation kit for Illumina (New England Biolabs, E7465).

## iPOND-MNase-seq analysis

iPOND-MNase-seq was performed as described[54] with modifications. EdU labeling, cross-linking, nucleosome-fragmentation, click reaction, and capture of streptavidin-coupled magnetic beads were performed as described above. Beads were washed four times in 1× B&W buffer (5 mM Tris-HCl pH 7.5, 1 M NaCl, 0.5 mM EDTA) and then libraries were prepared on beads using NEBNext Ultra II DNA library preparation kit for Illumina (New England Biolabs, E7465).

## MNase-seq data analysis

Sequenced reads were aligned to mm9 using bowtie2[66] and all uniquely matching reads were retained, using the following parameters. -I 0 -X 300 −no-mixed −no-discordant −very-sensitive. Average profiles were centered on TSS and calculated using qProfile command of QuasR[74]. Normalized average depth was calculated by (Counts of reads/Numbers of binding cites)/(Total reads/Genome size).

## Reporting summary

Further information on research design is available in the Nature Portfolio Reporting Summary linked to this article.

## Data availability

The data that support this study are available from the corresponding authors upon reasonable request. The high-throughput sequencing data generated in this study have been deposited in the GEO database under accession code GSE141560. The proteomic data used in this study are available in the Pride database under accession code PXD036330 Publicly available data utilized in this study are: RNA-seq for hematopoietic progenitor cells: GSE116177; Single cell RNA-seq profiling of HPSC: GSE81682 ; ChIP-seq for KLF4: GSM1324615; ChIP-seq for MYC: GSM912934; ChIP-seq for E2A: GSM546535; ChIP-seq for PAX5: GSM2863171; Mouse reference genome mm9: https://genome.ucsc.edu/cgi-bin/hgGateway?db=mm9. Source data are provided with this paper.

## Code availability

Custom codes used in this study are available via Zenode under DOI 10.5281/zenodo.7114888.

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

## Acknowledgements

We are grateful to K. Moro and S. Koyasu for supporting this project; V. J. Bardwell for *Bcor*$^{\Delta E9-10}$ mice; P. Y. Agata for retrovirus construct. This work was supported in part by Grants-in-Aid for Scientific Research from the Ministry of Education, Culture, Sports, Science and Technology of Japan (23249015 to H.K.), the Japan Agency for Medical Research and Development (AMED-CREST) (13417643 to H.K., S.I.), grants from the Japan Society for the Promotion of Science (18K19569 and 17H04214 to T.I.), RIKEN IMS Young Chief Investigator program (T.I.), The Mochida Memorial Foundation for Medical and Pharmaceutical Research (T.I.), and RIKEN Junior Research Associate Program (J.T.).

## Author contributions

Conceptualization: J.T., H.K., and T.I. Investigation: J.T., S.I., D.Y., Y.N.-T., T.U., J.S., Y.-W.H., K.I., Y.K., M.I., M.N., O.O., Y.H., E.A., R.K., and A.I. Bioinformatics: J.T., T.U, K.H., and E.A. Resources: S.I., T.K., Y.I., Y.K., M.S., and R.J.K. Writing: J.T., J.S., H.K. and T.I., Writing - review & editing: J.S., Supervision: H.K. and T.I., Funding acquisition: H.K. and T.I.

## Competing interests

The authors declare no competing interests.
