## [Peer Review File · Nature Communications]

REVIEWER COMMENTS

Reviewer #1 (Remarks to the Author):

Establishment of transcriptional repressed domains by polycomb group proteins is an extremely complicated field that remains an active area of investigation. One source of complexity is due to the presence of at least 6 forms of the polycomb repressive complex 1 (PRC1) that contain different isoforms of the Pcgf subunit (Pcgf1-6), as well as other subunits. Each form of PRC1 contains a Ring1a/b subunit that catalyzes monoubiquitinylation of histone H2A-K119 that facilitates recruitment of the PRC2 complex which can catalyze H3-K27me3 via its EZH2 subunit. In some cases, PRC2 action can lead to further recruitment of PRC1 complexes via recognition of histone methylation mark. Typically, inactivation of PRC1 complexes (Ring1b^{-/-}) or PRC2 (Ezh2^{-/-}) leads to upregulation of the same set of target genes, consistent with their recruitment interplay.

In this manuscript, the authors investigate the functional roles of Pcgf1 in hematopoietic stem and progenitor cells (HSPCs). They show that loss of Pcgf1 in a bone marrow transplantation model leads to abnormal, promiscuous differentiation along the myeloid lineage. As noted by the authors (ref 17-20), depletion of other subunits (e.g. Bcor or Kdm2b) of the Pcgf1-PRC1 complex yields a similar phenotype. The authors then use an immortalized HPC cell line (IdHPC) to investigate how loss of Pcgf1 impacts histone marks, PRC2 recruitment, and mRNA levels. First they show that Pcgf1 and Ring1b mark a set of genes (C1 cluster) that also have high levels of H3-K27me3 and PRC2 components. Expression of this cluster of genes is also de-repressed when Pcgf1 is depleted. In contrast, Pcgf1 and Ring1b marks a second set of genes (C2 cluster) that do not contain H3-K27me3, and their expression is not changed by Pcgf1 depletion. The functional significance of this data is not clear. In the absence of Pcgf1, the authors find that H3-K27me3 levels are reduced at C1 genes, but binding of subunits of PRC2 are not altered. The function of PRC2 is clearly important, as inactivation of EZH2 histone methylation also leads to de-repression of C1 gene expression.

As loss of Pcgf1 did not affect recruitment of PRC2 components, the authors hypothesize that the Pcgf1-PRC1 complex may function by a unique mechanism (i.e. function not by promoting PRC2 recruitment via H2A-ub). They show that a Pcgf1 weakly co-IPs the MCM helicase, as well as several abundant histone chaperones. They then use iPOND and a PLA assay to show that Pcgf1 is associated with replicating DNA. iPOND analysis from the Pcgf1^{-/-} cells shows increases in transcriptional regulators, and Mnase-iPOND shows decreases in levels of promoter proximal nucleosomes at both C1 and C2 cluster genes. These data lead the authors to suggest that Pcgf1-PRC1 functions at replication forks to promote nucleosome assembly, thus helping to restrict binding of transcription factors to target genes.

As one might surmise from this long initial summary, this is a complex paper with many seemingly separate stories combined into one. One solid, complete story demonstrates that Pcgf1 plays a key role in normal differentiation of HSPCs, much like is already known for the Bcor and Kdm2b subunits of this complex. The second story looks at the impact of Pcgf1 loss on PRC2 recruitment/activity and gene expression in immortalized HPCs. This story is incomplete (see below). Third, the role of Pcgf1 at replication forks is only tenuously connected to the first two stories, and it is also very incomplete. Their final conclusion: "Our findings, therefore, show that PRC1 and PRC2 are linked via previously unknown mechanisms in HSPCs to maintain proper differentiation potential and that such links are closely associated with the replication machinery." Is not supported by the data.

Major points:

1. Throughout the paper, the authors conclude that knockdown of Pcgf1 is equivalent to loss of Pcgf1-PRC1. This is certainly not the case, as the Ring1b subunit is still associated with target genes, as well as the H2AK119ub1 mark. Thus, these studies may inform on specific functions of Pcgf1, but not this noncanonical PRC1 complex.

2. In general, the authors' data support a canonical pathway for polycomb recruitment and function. A PRC1 complex helps to recruit PRC2, which together can repress target genes or recruit additional PRC1 complexes. The authors do a poor job at eliminating this simple model. Indeed, loss of Pcgf1 does not lead to loss of PRC2 components, but depletion also does not lead to loss of the Ring1b subunit of PRC1 or H2A-ub. Thus, this tells us something about Pcgf1, but does not eliminate simple models. Indeed, Figure 3d shows PRC2 function (EZH2) is required to repress C1 genes, much like the role of Pcgf1. ChIP and RNA expression analyses should be performed in parallel for Bcor or Kdm2b deletion cells, along with Ring1b^{-/-} cells. It was also odd that the authors did not probe the functioning of canonical PRC1 in their system (Pcgf2/4), as their model suggests that it may play no role.

3. A simple interpretation of the authors' results, that is not discussed, is that Pcgf1 functions to enhance the methylation activity of EZH2 or that it prevents recruitment/activity of a demethylase.

4. The authors do a poor job at referencing and discussing evidence that polycomb complexes associate with replication forks. Reference #12 is not appropriate, as this was an entirely *in vitro* study and does not show recruitment. The authors should cite Nat. Cell Biol. 10, 1291–1300 (2008) and Nat Commun 2014 Apr 14;5:3649.doi: 10.1038/ncomms4649. Latter paper indicates that PRC2 (EZH2) controls PRC1 requirement (Ring1b) at forks in MEFs.

5. The authors show that Pcgf1 is associated with nascent DNA, but this is not evidence that Pcgf1 actually functions at the fork. Detection by iPOND but would be consistent with rapid re-formation of polycomb domains after fork passage. To test this hypothesis correctly, you would need a means to disrupt fork association without impacting recruitment by other mechanisms.

6. The authors observe small changes in nucleosome density at both C1 and C2 clusters in the absence of Pcgf1. This would imply that these changes are not functionally important, as Pcgf1 has no impact on C2 gene expression. This needs a better discussion.

7. In contrast to nucleosome density, loss of H2A.Z is found specifically at C1 genes. A simple model is that this is due simply to increased transcription.

8. In many cases, changes in gene expression, nucleosome occupancy, or ChIP signals appear to be statistically significant, but the effects are often extremely small, making this reviewer concerned that much of this data is not biologically significant.

Other specific points:

Introduction, line 108. What is the evidence that Pcgf1 does not associate with other complexes? At best, a reference needs to be included here.

Figure S1d – why does Pcgf1 expression only reduce 3x in the deletion cells?

Figure 2a and Figure S2b. The authors find a disconnect between localization of

Pcgf1/Ring1b and H3-K27me3 in idHPCs. They compare their findings to previously published work in ESCs. This comparison would be more convincing if the authors had their own dataset, with samples treated identically.

The H2AK119ub1 ChIP data is very weak in Fig2 and Fig3. Not convincing signals. Same is true for Phc2 ChIPs.

Figure S3c. Why are all ChIPs not shown for Ring1b-/-? This seems key to distinguish impact of PRC1 compared to just Pcgf1. Also see no impact on EZH1 ChIP w/o Pcgf1, even though H3-K27me3 levels are low?

Figure 4g. Expression changes at Hmga2 are key for many of the conclusions of this paper. The authors need to show quantified levels, as was done in Figure 3F.

In many cases, the authors refer to their datasets as "biological Duplicates". Do they mean independent biological samples or replicas of a single biological sample?

Figure S5c. These co-IP results with replication fork factors are not convincing.

Figure 5c. The authors need to show each individual iPOND datasets, not just the subtracted set.

Figure 5A. Were these immunoprecipitation reactions performed as for the western blot analyses? What was the negative control sample here? Here they state in the methods that:

"Moreover, proteins and peptides identified as contaminating proteins were excluded. The global false discovery rate for both peptides and proteins was lower than 1% in this study".

How were contaminating proteins identified? The authors should provide the dataset, including number of peptides identified for each protein. Apparently, some proteins were only identified by one peptide obtained. It seems striking that histones were nearly equally significant as the replication factors, suggesting that this is the level of nonspecific chromatin binding.

Figure S5A. Here, the experiment should have been to use ChIP to follow these other subunits, not co-IP

In several places, the authors note that RUVL2 is a subunit of the hINO80 complex. But, it is also a subunit of the p400/Tip60 and SRCAP complexes that deposit H2A.Z. It is also in other complexes not linked to chromatin.

Reviewer #2 (Remarks to the Author):

The authors are aiming to shed light on the process of gene regulation in hematopoietic stem cells. It is known that non-canonical PRC1 complex (PRC1.1) plays a crucial role e.g. by repressing differentiation genes, yet the mechanism is unclear. In order to shed light on it, models conditional knock-out of the PRC1.1 component, PCGF1, in primary hematopoietic stem and progenitor cells (LSKs) and immortalized cell line, were used.

The KO mice had a defect in the hematopoietic system, with overall lower cell number in the bone marrow and accumulation of GMPs, suggesting myeloid bias in the differentiation. Using scRNAseq, the authors showed clear pattern of increased myeloid differentiation. To understand the mechanism of action of PRC1.1, ChIPseqs of PRC1, PRC2, histone marks

were performed. Based on the presence or absence of marks/components, 3 clusters of genes were identified: cluster 1 (PCGF1, RING1b, H3K27me3), cluster 2(PCGF1, RING1b) and cluster 3 (rest). In contrast to cluster 1, occupied by PRC2, cluster 1 was deprived of both PRC2 and H3K27me3, but was occupied by RNAP2 and H3K27ac mark. Upon depletion of PCGF1, only genes in C1 changed expression (upregulation), suggesting that those are in fact PRC1.1 targets. Strikingly, PCGF1 KO did not affect neither RING1b nor PRC2 occupancy, but caused a significant decrease of H3K27me3 mark. Gene ontology analysis revealed that cluster 1 contains stemness and myeloid differentiation genes. By using an EZH1/2 inhibitor (UNC1999) in wt cells, the same effect (upregulation of cluster 1 genes) was achieved, suggesting PRC1.1 function upstream of PRC2.

Immunoprecipitation experiments revealed interaction of PCGF1 with proteins related to replication and DNA organization, as well as chromatin remodelers. To validate possible role of PRC1.1 in replication, a proximity-ligation assay was performed, showing indeed the occupancy of PCGF1 on nascent DNAs. Furthermore, by using isolation of Proteins on Nascent DNA (iPOND) the interaction with remodelers was confirmed. iPOND coupled with MNase-seq further revealed a reduction of nucleosome occupancy at the +1 nucleosome in C1 and C2 genes, suggesting a role of PRC1.1 in regulating nucleosome density and by this mechanism regulating PRC2 activity. More in-depth study suggests that PRC1.1 regulates the deposition of H2A.Z nucleosome.

Although very insightful and informative, the article lacks some information as pointed out below:

- 1) It is unclear how the gates were selected in panel 1e, and if the % of cells represents the total cells analyzed (which should sum to 100%).
- 2) Are the data plotted in Figure 2g normalized by the total number of cells? If not, then it is difficult to appreciate the differences if the number of cells for the Control samples are lower than the cells analyzed for Pcgf1 KO.
- 3) What exactly is the cluster3 in Figure 2? Is this the rest of the genes, since in the same figure, the authors reported the value of 19.446. Yet, that would be quite surprising since H3K27ac decorates all of those regions.
- 4) It is important to discuss that PHC2 occupies the 481 genes of cluster1. This indicates a cross-talk between cPRC1 and ncPRC1 at those genes. For this, the authors should investigate if deletion of cPRC1 subunits would also trigger a de-repression of cluster1's genes.
- 4) Does deletion of Ring1A/B also affect expression of myeloid and stemness genes?
- 5) Treatment with EZH1/2 inhibitor phenocopies the deletion of Pcgf1 without affecting the H2Aub levels. Since the cross talk between ncPRC1 and PRC2 is mediated by H2Aub, it is unclear to the referee which is shared mechanism between these two sets of experiments.
- 5) Does overexpression of Hmg2a recapitulates Pcgf1 KO?
- 6) In supplementary figure 5a the authors should include Western blot analysis for PCLs proteins.
- 7) In order to claim a direct link between PCGF1 and replication machinery, the authors should perform CoIP experiments (in the presence of benzonase), eventually by adding and endogenous tag to PCGF1 for proper IP.
- 8) It would be interesting to explore the cluster 2, which although is occupied by PRC1.1, does not contain PRC2, changes nucleosome density at TSS upon PCGF1 KO, but seems not to be affected in terms of gene expression.

Reviewer #3 (Remarks to the Author):

Manuscript by Takano et al entitled "PCGF1-PRC1 safeguards recovery of nucleosome

positioning during S-phase to ensure early hematopoiesis". This is interesting work in which the authors uncover a novel role for the non-canonical PRC1 complex. The authors generate a conditional *Pcgf1* KO mouse model to study the role of PCGF1-PRC1 during early hematopoiesis and find that loss of *Pcgf1* results in myeloid skewing and suppresses lymphopoiesis. Cre-ERT2:*Pcgf1*^{fl/fl} BM cells are transplanted into wt recipient mice and tamoxifen-induced knockout of PCGF11 resulted in reduced BM and spleen cell numbers, a loss of B cell populations but an increase in GMP cells. HSC numbers did not change, but PCGF11 loss results in a change in lineage commitment towards myelopoiesis. These data were independently confirmed by scRNaseq. PCGF1 binds and downregulates myeloid genes, although loss of PCGF1 does not result in loss of PRC2 binding to these myeloid loci. Next, using a proteomics approach, the authors show that PCGF1 interacts with several proteins associated with the replication machinery, and the PCGF1 localises at the replication fork where it prevents excessive loading of transcriptional activators on nascent DNA. The authors identify that PCGF1-PRC1 facilitates nucleosome deposition immediately after the passage of replication fork, which is required for inheritance of proper chromatin conformation followed by DNA replication. The model that emerges is that at c1 loci that have H3K27me₃, PCGF1 is needed to prevent overloading of chromatin remodelers. In the absence of PCGF1, this does happen, and as a consequence PRC2 activity is reduced and repressive H3K27me₃ marks are lost. This occurs particularly at myeloid genes, and therefore after loss of PCGF1 cells undergo myeloid skewing. The authors focus mostly on c1 loci, but these are in fact a minority since c2 loci are more abundant, and since these loci are not in a repressed state (no repressive H3K27me₃ marks, but do carry RING1B, H3K27ac, RNAPII) one wonders whether at those loci *Pcgf1*/non canonical PRC1 would fulfill similar functions, most likely not. These are intriguing findings, but I do have a number of comments.

1. "Therefore, the interplay between PCGF1-PRC1 and cPRC1 observed in ESCs does not take place during normal or pathological hematopoiesis, indicating the presence of yet unknown mechanisms by which PCGF1-PRC1 mediates gene repression." It remains unclear why the role of PCGF1-PRC1 would be different in ES cells compared to HSCs, also at the mechanistic level. While in ES cells there appears to be a strong overlap between PRC2 and PRC1 occupied loci that often also carry H3K27me₃, in HSCs and LSCs this is clearly different. The authors themselves also show that the majority of PCGF1-bound loci are in fact in an active chromatin conformation (fig.2, c2 loci, 815 loci bound by PCGF11, RING1B, H2K27ac, RNAPII) while less loci are in a repressed conformation (481 c1 loci, PCGF1, PRC2, H3K27me₃). Yet, the majority of the story focuses on the role of PCGF1 at these c1 loci, whereby loss of PCGF1 results in overloading of chromatin remodelers at the replication fork, which in turn negatively impacts on PRC2 activity and as a consequence a reduction in repressive H3K27me₃ marks is seen, resulting in enhanced myeloid gene expression. The function of PCGF1 at the active loci is then different since there is no PRC2 at all which activity therefore also does not need to be controlled by preventing overloading of chromatin remodelers. This is not addressed.

2. The authors appear to ignore somewhat the potential role of non-canonical PRC1 at active loci, not only by focusing mostly on the repressed c1 loci but also in their discussion of the available literature. The authors also exclusively focus on its tumor suppressive roles while various papers have shown that non-canonical PRC1/KDM2B can also act as an oncogene (Andricovich et al, 2016; He et al, 2011; Kottakis et al, 2014; Ueda et al, 2015; van den Boom et al., 2016). These papers highlight a different role and should be discussed.

3. Suppl Fig.1D: why is PCGF1 expression not completely gone? This reduction seems rather modest.

4. Figure 1g: why could C3, C6, C7, and C8 clusters not be annotated? Was there a problem with sequence depth? How many transcripts were in fact quantified? In the pseudo-time differentiation trajectories depicted in fig 1i: why are the TMPs localized further away from the root state/HSCs compared to GMPs? One would expect that TMPs would precede GMPs? Also MPPs locate in a separate branch, why?

5. The use of ID3 overexpressing immortalized multipotent progenitor lines is interesting. The authors did not use the inducible model they published previously. Are the authors sure that ID3 does not interfere with non-canonical PRC1 chromatin binding characteristics? In figure 4 primary LMPP or LSK cells were isolated from BM of ERT2-Cre (control) and ERT2-Cre:Pgcf1fl/fl (Pgcf1-KO) mice that were cultured for 4 days in the presence of 4-OHT. These do not express ID3, which would serve as a good control. What I do not understand is why the authors only focus on the 481 c1 loci and 815 c2 loci that were identified in the IdHPCs, why not simply show which genes were affected in these primary HSPCs and then overlay those with what was seen in IdHPCs? Please provide these comparisons as well.

6. The effects of Pgcf1 KO on gene expression changes seems relatively small, even in the c1 subgroup of repressed loci. Are all genes depicted in suppl fig 1e with a \log_2 FC > 1.5 also statistically significant? Same for genes plotted in fig 2e. How many genes are upregulated in suppl fig 1e, and what is the overlap with the c1-c2-c3 subsets in fig 2a-b? Similar for the total up/down genes depicted in suppl fig 2d. Please validate these gene expression changes by independent Q-PCRs/Westerns. HMGA2 is picked as a candidate for further validations, but only changes of chromatin marks are shown, not the effects on gene expression changes, only fig 4g showing genome browser tracks of the expression of Hmga2 in control and Pgcf1-KO LMPPs, which only appears to show modest effects. And is only 1 exon shown? Please clarify. The effects of the by the EZH1/2 inhibitor UNC1999 are clear on HMGA2 expression (Fig. 3f), but this is not the same as PCGF1 KO. Cluster 2 not only contains B-cell genes, but the strongest group is embryonic organ development (Fig. 2d). What is the function of PCGF1 at these loci? It is very clear that 2 distinct non-canonical PRC1 chromatin states exist in HSPCs, repressed and active states. It remains puzzling what mechanisms control these 2 different states, and what the role of PCGF1 at the active loci is. The authors focus exclusively on the role of PCGF1 at repressed loci and do not discuss at all the potential role of ncPRC1 at active foci.

7. Is it not surprising that the shHmga2 -1 with a rather modest 50% knockdown efficiency (Suppl fig. 4c) almost completely restores B cell numbers/%s (fig. 4 j-k)? Although I do agree that it is of interest to see that overexpression of Hma2 in TET2 deficient background appears to be sufficient to drive myeloid transformation (Bai et al 2021).

8. Whether the enzymatic activity of PRC2 is truly affected by overloading of chromatin remodelers is formally not shown, the authors refer to 2 papers (refs 22-23) but do not provide data in their model systems in the absence of PCGF1.

9. Although experimentally probably difficult to address: is the interactome between PCGF1 and DNA replication machinery mostly derived from PCGF1 at c1 repressed loci? In fact the majority of PCGF1 peaks do not appear to be in the repressed c1 loci, and therefore the interactome would not only reflect the situation at c1 loci, but also at active c2 loci. One would presume that loss of PCGF1 also results in an overloading of chromatin remodelers at active c2 loci, what would be the consequence of that? Or rather, why would there not be consequences for those loci, the authors only show that expression of these loci does not appear to be altered (eg further increased over basal levels that are already higher). It would be interesting to have the authors' thoughts on this.

Minor:

Line 100: "...PCGF4 facilitates maintenance of leukemic stem cells..." please also cite papers focusing on human leukemias (DOI: 10.1182/blood-2009-03-209734; doi.org/10.1182/blood-2010-02-270660)

Typo in line 195 These cells, therefore, can be utilized as a substitute for LMPPs for.

Line 198 "We found that PCGF1 was efficiently depleted in IdHPCs after 4 days of 4-OHT injection (Supplementary Fig. 1l)" Injection suggests in vivo but assume 4OHT was administered in vitro?

The -/+ 10kb from TSS indicator at the bottom is mispositioned.

C3=20666 genes, in the VENN diagram in fig.2A it is not immediately clear where this number comes from.

Maybe add fold changes to Suppl Fig 2G, fig.2C, Fig.3B for clarity

In the discussion, paragraph starting at line 404, please include previous work in the discussion that showed the existence of active loci bound/controlled by non-canonical PRC1 (DOI: 10.1016/j.celrep.2015.12.034)

Point-by-point reply to Reviewer comments:

Reviewer #1 (Remarks to the Author):

Establishment of transcriptional repressed domains by polycomb group proteins is an extremely complicated field that remains an active area of investigation. One source of complexity is due to the presence of at least 6 forms of the polycomb repressive complex 1 (PRC1) that contain different isoforms of the Pcgf subunit (Pcgf1-6), as well as other subunits. Each form of PRC1 contains a Ring1a/b subunit that catalyzes monoubiquitinylation of histone H2A-K119 that facilitates recruitment of the PRC2 complex which can catalyze H3-K27me3 via its EZH2 subunit. In some cases, PRC2 action can lead to further recruitment of PRC1 complexes via recognition of histone methylation mark. Typically, inactivation of PRC1 complexes (Ring1b^{-/-}) or PRC2 (Ezh2^{-/-}) leads to upregulation of the same set of target genes, consistent with their recruitment interplay.

In this manuscript, the authors investigate the functional roles of Pcgf1 in hematopoietic stem and progenitor cells (HSPCs). They show that loss of Pcgf1 in a bone marrow transplantation model leads to abnormal, promiscuous differentiation along the myeloid lineage. As noted by the authors (ref 17-20), depletion of other subunits (e.g. Bcor or Kdm2b) of the Pcgf1-PRC1 complex yields a similar phenotype. The authors then use an immortalized HPC cell line (IdHPC) to investigate how loss of Pcgf1 impacts histone marks, PRC2 recruitment, and mRNA levels. First they show that Pcgf1 and Ring1b mark a set of genes (C1 cluster) that also have high levels of H3-K27me3 and PRC2 components. Expression of this cluster of genes is also de-repressed when Pcg1 is depleted. In contrast, Pcgf1 and Ring1b marks a second set of genes (C2 cluster) that do not contain H3-K27me3, and their expression is not changed by Pcg1 depletion. The functional significance of this data is not clear. In the absence of Pcg1, the authors find that H3-K27me3 levels are reduced at C1 genes, but binding of subunits of PRC2 are not altered. The function of PRC2 is clearly important, as inactivation of EZH2 histone methylation also leads to de-repression of C1 gene expression.

As loss of Pcgf1 did not affect recruitment of PRC2 components, the authors hypothesize that the Pcgf1-PRC1 complex may function by a unique mechanism (i.e. function not by promoting PRC2 recruitment via H2A-ub). They show that a Pcgf1

weakly co-IPs the MCM helicase, as well as several abundant histone chaperones. They then use iPOND and a PLA assay to show that Pcgf1 is associated with replicating DNA. iPOND analysis from the Pcgf1^{-/-} cells shows increases in transcriptional regulators, and Mnase-iPOND shows decreases in levels of promoter proximal nucleosomes at both C1 and C2 cluster genes. These data lead the authors to suggest that Pcgf1-PRC1 functions at replication forks to promote nucleosome assembly, thus helping to restrict binding of transcription factors to target genes.

As one might surmise from this long initial summary, this is a complex paper with many seemingly separate stories combined into one. One solid, complete story demonstrates that Pcgf1 plays a key role in normal differentiation of HSPCs, much like is already known for the Bcor and Kdm2b subunits of this complex. The second story looks at the impact of Pcgf1 loss on PRC2 recruitment/activity and gene expression in immortalized HPCs. This story is incomplete (see below). Third, the role of Pcgf1 at replication forks is only tenuously connected to the first two stories, and it is also very incomplete. Their final conclusion: “Our findings, therefore, show that PRC1 and PRC2 are linked via previously unknown mechanisms in HSPCs to maintain proper differentiation potential and that such links are closely associated with the replication machinery.” Is not supported by the data.

Major points:

1. Throughout the paper, the authors conclude that knockdown of Pcgf1 is equivalent to loss of Pcgf1-PRC1. This is certainly not the case, as the Ring1b subunit is still associated with target genes, as well as the H2AK119ub1 mark. Thus, these studies may inform on specific functions of Pcgf1, but not this noncanonical PRC1 complex.

Our Reply)

We appreciate this comment. We have addressed this issue by comparing molecular changes in *Pcgf1*-KO IdHPCs with those in *Bcor* (encodes another component of PCGF1-PRC1) mutant (*Bcor*^{Δ⁹⁻¹⁰}), *Pcgf2/4* (components of canonical PRC1) mutant and *Ring1a/b*-dKO (encode catalytic components of PRC1) IdHPCs. We observed up-regulation of Cluster 1 (C1) genes in *Bcor*^{Δ⁹⁻¹⁰}, *Pcgf1*-KO and, to a higher level, in *Ring1a/b*-dKO IdHPCs but not in *Pcgf2/4*-dKO (Fig. 2c, f, g, Supplementary Fig. 2i, j, k, l). We further analyzed local distribution of RING1B, H2AK119ub1, PRC2 H3K27me3, and cPRC1 in *Pcgf1*-KO, *Bcor*^{Δ^{E9-10}}, *Pcgf2/4*-dKO and *Ring1a/b*-dKO IdHPCs. In *Pcgf1*-KO and *Bcor*^{Δ^{E9-10}} IdHPCs, reduction of H3K27me3 does not accompany a decrease of H2AK119ub1 marks, PRC2 or cPRC1 binding (Fig. 3d, Supplementary Fig. 3a-f). In

contrast, H3K27me3 and H2AK119ub1 levels were unaltered despite considerable reduction of RING1B binding in *Pcgf2/4*-dKO IdHPCs, suggesting RING1A/B incorporated in cPRC1 is dispensable for H3K27me3 deposition. Importantly, in *Ring1a/b*-dKO IdHPCs, reduction of H3K27me3 was accompanied by decrease of H2AK119ub1, PRC2 and cPRC1 at C1 genes. Therefore, up-regulation of the C1 genes and reduction of H3K27me3 at the C1 were seen in *Pcgf1*-KO, *Bcor*^{Δ9-10} and *Ring1a/b*-dKO IdHPCs but not in *Pcgf2/4*-dKO. These observations support that PCGF1 functions as a functional component of PCGF1-PRC1 in IdHPCs as well as BCOR. It is also notable that RING1A/B incorporated into other variant PRC1, such as PCGF3/5/6-PRC1, was suggested to facilitate H3K27me3 deposition by activating H2AK119ub1/PRC2 pathway in parallel with PCGF1-PRC1-dependent pathway. Indeed, hematopoietic phenotypes of *Pcgf1*-KO were similar to those of *Bcor* mutant and *Kdm2b*-KO (refs. 31-34). In addition, our IP-MS study revealed dominant association of 3xFLAG-PCGF1 with RING1B (RNF2), RING1A (RING1), KDM2B, and BCORL1 but not components of other variant or canonical PRC1 except for RING1A/B (Fig. 5a). Although we found association of 3xPCGF1 to DNA helicase complexes, remodeling factors, and histone chaperones, these interactions were substoichiometric and expected to represent PCGF1-association on nascent DNA that appear upon replication fork passage (Supplementary Fig. 5d, f). All these observations support that PCGF1 functions as a functional component of PCGF1-PRC1 but not other variant or canonical PRC1 in hematopoietic lineage.

Moreover, target binding of PCGF1 was reported to depend on RING1A/B in previous study (ref. 37). This clearly demonstrates PCGF1 binds to target genes as a component of PRC1 in ESCs. These additions are described in lines 255-272, 288-305, 374-378 and 120-122 of the main text of the revised version.

2. In general, the authors' data support a canonical pathway for polycomb recruitment and function. A PRC1 complex helps to recruit PRC2, which together can repress target genes or recruit additional PRC1 complexes. The authors do a poor job at eliminating this simple model. Indeed, loss of *Pcgf1* does not lead to loss of PRC2 components, but depletion also does not lead to loss of the *Ring1b* subunit of PRC1 or H2A-ub. Thus, this tells us something about *Pcgf1*, but does not eliminate simple models. Indeed, Figure 3d shows PRC2 function (EZH2) is required to repress C1 genes, much like the role of *Pcgf1*. ChIP and RNA expression analyses should be performed in parallel for *Bcor* or *Kdm2b* deletion cells, along with *Ring1b*^{-/-} cells. It was also odd that the authors did not probe the functioning of canonical PRC1 in their system (*Pcgf2/4*), as their model suggests that it may play no role.

Our reply)

We appreciate this suggestion. According to this suggestion, we have newly generated *Bcor*^{Δ9-10} and *Pcgf2/4*-dKO IdHPCs and comparatively investigated gene expression changes and H3K27me3 depositions among *Pcgf1*-KO, *Bcor*^{Δ9-10}, *Pcgf2/4*-dKO and *Ring1a/b*-dKO IdHPCs as mentioned in our reply to the comment 1 made by the reviewer #1. These results are shown in Fig. 2f, 2g, 2h, supplementary Fig. 2l, Fig. 3d, and supplementary Fig. 3a, 3b, 3c, 3d, and 3e in the revised version. In summary, we found that PCGF1-PRC1 regulates H3K27me3 level at C1 genes in an H2AK119ub1-independent manner. This non-canonical pathway was revealed to cooperate with the H2AK119ub1/PRC2 pathway, likely downstream of other variant PRC1 sub-complexes harboring PCGF3/5/6 but not PCGF1. In contrast, cPRC1 sub-complexes appear to be dispensable during this process. These additions are described in lines 255-272 and 288-305 of the main text of the revised version.

3. A simple interpretation of the authors' results, that is not discussed, is that Pcgf1 functions to enhance the methylation activity of EZH2 or that it prevents recruitment/activity of a demethylase.

Our reply)

We appreciate this comment. We examined target binding and complex formation of PRC2.1 and PRC2.2 and found that they were normally formed and bind to C1 genes in *Pcgf1*-KO IdHPCs (Fig. 5a, Supplementary Fig. 5b, c). We also asked whether demethylase activity for H3K27me3 could be more abundantly accumulated in the absence of PCGF1. We, however, observed that enrichment of UTX around TSS was not significantly changed in *Pcgf1*-KO (supplementary Fig. 5c). These observations prompted us to suspect that PCGF1-PRC1 affects PRC2 activity in an indirect manner. Consistent with this model, we observed that SMARCA4 inhibitor (ACB11) could concurrently restored nucleosome density and H3K27me3 deposition at the C1 genes in *Pcgf1*-KO IdHPCs to some extents. As nucleosome density is accepted to regulate PRC2 activity (ref. 36), we expect PCGF1-PRC1 regulates PRC2 activity by antagonizing SMARCA4-mediated nucleosome remodeling. These results and discussion are described in lines 364-381 and 406-438 of the revised main text.

4. The authors do a poor job at referencing and discussing evidence that polycomb complexes associate with replication forks. Reference #12 is not appropriate, as this was an entirely in vitro study and does not show recruitment. The authors should cite Nat. Cell Biol. 10, 1291?1300 (2008) and Nat Commun 2014 Apr 14;5:3649.doi: 10.1038/ncomms4649. Latter paper indicates that PRC2 (EZH2) controls PRC1 requirement (Ring1b) at forks in MEFs.

Our reply)

We appreciate this suggestion. We have added these papers as refs. 21 and 22 in the revised version in lines 92-93.

5. The authors show that *Pcgf1* is associated with nascent DNA, but this is not evidence that *Pcgf1* actually functions at the fork. Detection by iPOND but would be consistent with rapid re-formation of polycomb domains after fork passage. To test this hypothesis correctly, you would need a means to disrupt fork association without impacting recruitment by other mechanisms.

Our reply)

We appreciate this thoughtful comment. As we labeled IdHPCs for 20 minutes by EdU, EdU-labeled DNA should harbor certain heterogeneity. Nonetheless, we found considerable enrichment of PCNA, DNMT1 and UHRF1, which are functional components of replication machinery, on EdU-labeled nascent DNA in our experimental condition (Fig. 5d). This implies PCGF1 efficiently associate with the nascent DNA to the similar extents with replication machinery as revealed by PLA assay (Fig. 5b), iPOND-ChIP seq (Fig. 5c) and iPOND-MS (Fig. 5d). However, we should also share the reviewer #1's concern that this does not formally exclude the possibility of rapid re-formation of PCGF1-PRC1 after the passage of the replication fork. We therefore tried to disrupt PCGF1 at the fork by means of PIP-degron technique. We, however, found PIP-tagged PCGF1 failed to associate with BCOR and this experiment undoable at this moment. We therefore modified the text to state that PCGF1-PRC1 functions "on the nascent DNA" upon replication fork passage or "in the vicinity of replication fork" instead "at replication fork".

6. The authors observe small changes in nucleosome density at both C1 and C2 clusters in the absence of *Pcgf1*. This would imply that these changes are not functionally important, as *Pcgf1* has no impact on C2 gene expression. This needs a better discussion.

Our reply)

We appreciate this comment. To assess biological impacts of changes in nucleosome density in *Pcgf1*-KO IdHPCs, we first tested whether SMARCA4 inhibitor (ACB11) could restore reduced nucleosome density induced by PCGF1 depletion to examine whether restoration of nucleosome density could restore gene expression changes and H3K27me3 deposition at the C1 genes and *Hmga2* expression (a representative C1 gene) due to PCGF1 depletion. We indeed observed that ACB11 treatment restored nucleosome density and concurrently gene expression and H3K27me3 deposition in *Pcgf1*-KO IdHPCs (Fig. 5g, h, i, l). As nucleosome density is known to rate-limit PRC2 activity (ref. 36), SMARCA4-mediated remodeling induced by PCGF1

depletion could be a cue to down-regulate H3K27me3 deposition and subsequently up-regulate the C1 genes. These results are described in lines 415-438. Unlike the C1 genes, the C2 genes are more abundantly expressed than C1 genes and depleted by H3K27me3. Therefore, PRC2-mediated repressive pathway is not active at the C2 genes (Fig. 2b, 2c, Fig. 5i). Potential roles of PCGF1-PRC1 at C2 genes are discussed in lines 495-518.

7. In contrast to nucleosome density, loss of H2A.Z is found specifically at C1 genes. A simple model is that this is due simply to increased transcription.

Our reply)

Although we appreciate this comment, we have deleted data and argument related to H2A.Z in the revised version. This is because we could not show functional impacts of H2A.Z loss for either gene expression changes or H3K27me3 deposition in *Pcgfl*-KO. In contrast, SMARCA4-related pathway can at least partially explain how PCGF1 functions in association with replication fork passage. To make the whole story simpler and easy-reading, we made decision to delete H2A.Z-related description in the revised version. If the reviewer considers H2A.Z-related data critically important, we are happy to relocate them.

8. In many cases, changes in gene expression, nucleosome occupancy, or ChIP signals appear to be statistically significant, but the effects are often extremely small, making this reviewer concerned that much of this data is not biologically significant.

Our reply)

We appreciate this comment. We thus experimentally addressed this concern. We first optimized respective experimental conditions, repeated every experiment at least a few times and addressed the same issues in different ways to warrant reproducibility of respective analyses. RNA-seq and ChIP-seq data were respectively followed by confirmation by RT-qPCR and ChIP-qPCR, which allowed us to observe changes in more substantial manner (see Fig. 2g, 3c, 3d, 4c, 4g, 5i, 5j). Moreover, we assessed biological impacts of molecular changes by giving experimental perturbations such as treatment by SMARCA4 inhibitor and shRNA for *Hmga2*. As mentioned previously, we observed that SMARCA4 inhibitor (ACB11) could concurrently restored nucleosome density and H3K27me3 deposition at the C1 genes and *Hmga2* expression (a representative C1 gene) in *Pcgfl*-KO IdHPCs though the restoration was partial as described in lines 415-438 (Fig. 5g, h, i, j). shRNA for *Hmga2* was also shown to partially restore hematopoietic defects in *Pcgfl*-KO as described in lines 349-351 (Fig. 4h, 4i, 4j, Supplementary Fig. 4f, 4g). Taken together, though molecular changes seen in *Pcgfl*-KO were often small, they are highly reproducible.

Moreover, by comparing these changes in *Pcgf1*-KO IdHPCs with those in *Ring1a/b*-dKO and EZH1/2 inhibitor-treated IdHPCs, impacts of PCGF1-PRC1 to facilitate H3K27me3 deposition were revealed to be compensated by other variant PRC1 such as PCGF3/5/6-PRC1, which activate H2AK119ub1-dependent PRC2 recruitment. Presence of compensation may make molecular changes small. This possibility is discussed in lines 476-494.

Other specific points:

1. Introduction, line 108. What is the evidence that *Pcgf1* does not associate with other complexes? At best, a reference needs to be included here.

Our reply)

We appreciate this comment. We have modified the text and added a reference (ref.37), in which target binding of PCGF1 was shown to be RING1A/B-dependent. This is described in lines 120-122.

2. Figure S1d ? why does *Pcgf1* expression only reduce 3x in the deletion cells?

Our reply)

We appreciate this comment. We have replaced supplementary Fig. 1d with the data obtained from LMPPs instead of HSCs. In LMPPs, *Pcgf1* transcripts could not be detected as shown in supplementary Fig.1d. This is described in lines 151-153.

3. Figure 2a and Figure S2b. The authors find a disconnect between localization of *Pcgf1*/*Ring1b* and H3-K27me3 in idHPCs. They compare their findings to previously published work in ESCs. This comparison would be more convincing if the authors had their own dataset, with samples treated identically.

Our reply)

We appreciate this suggestion. To enable direct comparison between IdHPCs and ESCs, we showed ChIP-seq for RING1B, H2AK119ub1, SUZ12 and H3K27me3 in ESCs performed by our hands as shown in supplementary Fig. 2e and described in lines 229-232 in the revised version. We confirmed SUZ12 binding and H3K27me3 deposition to the C2 genes and impacts of PCGF1 depletion on H2AK119ub1 and H3K27me3 in ESCs, as shown in supplementary Fig. 3g, 3i and 3j. We found the enrichment of H3K27me3 over C2 genes as well as C1 genes and also destabilization of H2AK119ub1 upon *Pcgf1* deletion in ESCs. This is described in lines 285-287.

4. The H2AK119ub1 ChIP data is very weak in Fig2 and Fig3. Not convincing signals. Same is true for Phc2 ChIPs.

Our reply)

We appreciate this comment. Regarding the H2AK119ub1 ChIP, we have fine-tuned the protocol in which chromatin was more strictly fragmented by MNase. This procedure yielded clearer signals for both (Fig. 2b and Fig. 3a). We have replaced the figures accordingly and also provided control data by using *Ring1a/b*-dKO. For H2AK119ub1 signals, we found H2AK119ub1 signals fully disappeared in *Ring1a/b*-dKO IdHPCs (Fig. 3a). Similarly, we found PHC2 enrichment disappeared in *Pcgf2/4*-dKO IdHPCs (Please see the figure for Reviewer #1).

5. Figure S3c. Why are all ChIPs not shown for *Ring1b*-/-? This seems key to distinguish impact of PRC1 compared to just *Pcgf1*. Also see no impact on EZH1 ChIP w/o *Pcgf1*, even though H3-K27me3 levels are low?

Our reply)

We appreciate this comment. We performed ChIP-qPCR analysis for H2AK119ub1, RING1B, SUZ12, H3K27me3 and PHC2 in *Ring1a/b*-dKO IdHPCs in parallel with *Pcgf1*-KO, *Bcor*^{Δ9-10} and *Pcgf2/4*-dKO as shown in Fig. 3c, 3d, supplementary Fig. 3e and 3f, and came up with a more comprehensive picture how PCGF1 functions to regulate H3K27me3. In *Ring1a/b*-dKO IdHPCs, target bindings of H2AK119ub1, RING1B, SUZ12, H3K27me3 and PHC2 were obviously reduced. In contrast, in *Pcgf1*-KO and *Bcor*^{Δ9-10} IdHPCs, changes were only seen in H3K27me3 distribution but neither in H2AK119ub1, RING1B, SUZ12, or PHC2. In *Pcgf2/4*-dKO, we observed reduction in target bindings of RING1B and PHC2 but not in H2AK119ub1, RING1B, or H3K27me3 either. Therefore depositions of H2AK119ub1 and PRC2 are only affected in *Ring1a/b*-dKO IdHPCs. Although this implies that H2AK119ub1-dependent pathway to recruit PRC2 is active in IdHPCs, PCGF1 and BCOR are dispensable to activate this cascade. Similarly, RING1A/B loaded via their association with canonical PRC1 do not play a major role in H2AK119ub1/PRC2 pathway either. We thus expect this pathway could be driven mainly by variant PRC1 sub-complexes that incorporate PCGF3/5/6, as reported by Fursova et al. (Ref.46). These additions are described in lines 289-305 of the main text of the revised version.

6. Figure 4g. Expression changes at *Hmga2* are key for many of the conclusions of this paper. The authors need to show quantified levels, as was done in Figure 3F.

Our reply)

We appreciate this comment. We investigated the expression of *Hmga2* by using RT-qPCR in *Pcgf1*-KO, *Bcor*^{Δ9-10}, *Pcgf2/4*-dKO and *Ring1a/b*-dKO IdHPCs, and observed considerable up-regulation of *Hmga2* in *Pcgf1*-KO, *Bcor*^{Δ9-10} and *Ring1a/b*-dKO IdHPCs but not *Pcgf2/4*-dKO (Fig. 2g). Furthermore, up-regulation of HMGA2 protein in *Pcgf1*-KO IdHPCs was confirmed by IB analysis (Fig. 2h). We also observed up-regulation of *Hmga2* in *Pcgf1*-KO LMPPs (Fig. 4c). Treatment of *Pcgf1*-KO IdHPCs by SMARCA4 inhibitor was shown to induced down-regulation of *Hmga2* (Fig. 5j). These additions are described in lines 267-270 and lines 435-438 of the main text of revised version.

7. In many cases, the authors refer to their datasets as “biological Duplicates”. Do they mean independent biological samples or replicas of a single biological sample?

Our reply)

We appreciate this comment. For better clarification, we state this more precisely in respective figure legends.

8. Figure S5c. These co-IP results with replication fork factors are not convincing.

Our reply)

We appreciate this comment. To perform IP in more physiological condition, we have generated a new allele for *Pcgf1*, in which TY1-tagged PCGF1 is expressed (Extended Data Fig. 2), used IdHPCs expressing endogenously TY1-tagged PCGF1 for IP-immunoblotting (Supplementary Fig. 5f). Here, we observed weak signals for MCM7, PCNA and SMARCA4 compared to those for PCGF1 and RING1B but these results turned out to be reproducible. We speculate that this is due to transient and substoichiometric nature of interaction between PCGF1-PRC1 and replication machinery that occurs only during the fork passage. According to these additions, we have modified the main text in lines 385-386.

9. Figure 5c. The authors need to show each individual iPOND datasets, not just the subtracted set.

Our reply)

We appreciate this suggestion. Accordingly, we modified Fig. 5d to show all data that involve quantity information (Fig. 5d).

10. Figure 5A. Were these immunoprecipitation reactions performed as for the western blot analyses? What was the negative control sample here? Here they state in the methods that:

“Moreover, proteins and peptides identified as contaminating proteins were excluded. The global false discovery rate for both peptides and proteins was lower than 1% in this study”.

How were contaminating proteins identified? The authors should provide the dataset, including number of peptides identified for each protein. Apparently, some proteins were only identified by one peptide obtained. It seems striking that histones were nearly equally significant as the replication factors, suggesting that this is the level of nonspecific chromatin binding.

Our reply)

We appreciate these comments. Here, we used *Pcgfl*-KO IdHPCs expressing 3xFLAG-tagged “empty” vector as a negative control of IP-MS analysis. Regarding the contaminating proteins, we referred to the standard MaxQuant contaminants database (http://www.coxdocs.org/doku.php?id=maxquant:start_downloads.htm). In addition, as suggested by the Reviewer #1, we have deposited our data including number of peptides identified for each protein to the public database ProteomeXchange Consortium via the jPOST partner repository with the dataset identifier PXD033883 for ProteomeXchange and JPST001586 for jPOST.

In addition, we used an updated method to analyze MS data in the revised version. Technical details are described in the materials and method section as following; All MS/MS files were searched against the UniProtKB/Swiss-Prot mouse database (Proteome ID: UP000000589, downloaded October 30, 2019, 17069 proteins entries), combined with the standard MaxQuant contaminants database (http://www.coxdocs.org/doku.php?id=maxquant:start_downloads.htm), using ProteinPilot software v. 4.5 with Paragon algorithm for protein identification. The search parameters were as follows: cysteine alkylation of iodoacetamide, trypsin digestion, and TripleTOF 5600. For a protein confidence threshold, we used the ProteinPilot unused score of 1.3 with at least one peptide with 95% confidence. Moreover, proteins and peptides identified as contaminating proteins were excluded. Global false discovery rate for both peptides and proteins was lower than 1% in this study.

11. Figure S5A. Here, the experiment should have been to use ChIP to follow these other subunits, not co-IP.

Our reply)

We appreciate this comment. According to the suggestion, we performed ChIP-seq for KDM2B, BCOR, JARID2 (Fig. 3a) and PCL2 (Supplementary Fig. 5c). Accordingly, we modified the text in lines 278, 281 and 369-371.

12. In several places, the authors note that RUVL2 is a subunit of the hINO80 complex. But, it is also a subunit of the p400/Tip60 and SRCAP complexes that deposit H2A.Z. It is also in other complexes not linked to chromatin.

Our reply)

We appreciate this suggestion. As discussed in our reply for major comment 7 made by the reviewer #1, we have deleted data and description related to H2A.Z.

Reviewer #2 (Remarks to the Author):

The authors are aiming to shed light on the process of gene regulation in hematopoietic stem cells. It is known that non-canonical PRC1 complex (PRC1.1) plays a crucial role e.g. by repressing differentiation genes, yet the mechanism is unclear. In order to shed light on it, models conditional knock-out of the PRC1.1 component, PCGF1, in primary hematopoietic stem and progenitor cells (LSKs) and immortalized cell line, were used. The KO mice had a defect in the hematopoietic system, with overall lower cell number in the bone marrow and accumulation of GMPs, suggesting myeloid bias in the differentiation. Using scRNAseq, the authors showed clear pattern of increased myeloid differentiation.

To understand the mechanism of action of PRC1.1, ChIPseqs of PRC1, PRC2, histone marks were performed. Based on the presence or absence of marks/components, 3 clusters of genes were identified: cluster 1 (PCGF1, RING1b, H3K27me3), cluster 2(PCGF1, RING1b) and cluster 3 (rest). In contrast to cluster 1, occupied by PRC2, cluster 1 was deprived of both PRC2 and H3K27me3, but was occupied by RNAP2 and H3K27ac mark. Upon depletion of PCGF1, only genes in C1 changed expression (upregulation), suggesting that those are in fact PRC1.1 targets. Strikingly, PCGF1 KO did not affect neither RING1b nor PRC2 occupancy, but caused a significant decrease of H3K27me3 mark. Gene ontology analysis revealed that cluster 1 contains stemness and myeloid differentiation genes. By using an EZH1/2 inhibitor (UNC1999) in wt cells, the same effect (upregulation of cluster 1 genes) was achieved, suggesting PRC1.1 function upstream of PRC2.

Immunoprecipitation experiments revealed interaction of PCGF1 with proteins related to replication and DNA organization, as well as chromatin remodelers. To validate possible role of PRC1.1 in replication, a proximity-ligation assay was performed, showing indeed the occupancy of PCGF1 on nascent DNAs. Furthermore, by using

isolation of Proteins on Nascent DNA (iPOND) the interaction with remodelers was confirmed. iPOND coupled with MNase-seq further revealed a reduction of nucleosome occupancy at the +1 nucleosome in C1 and C2 genes, suggesting a role of PRC1.1 in regulating nucleosome density and by this mechanism regulating PRC2 activity. More in-depth study suggests that PRC1.1 regulates the deposition of H2A.Z nucleosome. Although very insightful and informative, the article lacks some information as pointed out below:

1) It is unclear how the gates were selected in panel 1e, and if the % of cells represents the total cells analyzed (which should sum to 100%).

Our reply)

We appreciate this comment. According to this comment, we described the gating strategy in Extended Data Fig. 1. Accordingly, we modified the text in lines 156-163.

2) Are the data plotted in Figure 2g normalized by the total number of cells? If not, then it is difficult to appreciate the differences if the number of cells for the Control samples are lower than the cells analyzed for *Pcgf1* KO.

Our reply)

We appreciate this comment. As Fig. 2g was not present in our previously submitted version, we suppose the Reviewer #2 is asking about Fig. 1g (since Fig. 3g was not present either). In Fig. 1g, we show the UMAP representation for scRNA-seq data of control and *Pcgf1*-KO LSK cells, in which each dot represents each cell. We summarized these data in Fig. 1f, in which frequency of cells in each cluster between control and *Pcgf1*-KO LSK cells.

3) What exactly is the cluster3 in Figure 2? Is this the rest of the genes, since in the same figure, the authors reported the value of 19.446. Yet, that would be quite surprising since H3K27ac decorates all of those regions.

Our reply)

We appreciate this comment. Yes, as Reviewer #2 indicated, the cluster 3 (C3) represented the rest of genes. According to the comment, we realized that more precise classification of genes needed and performed k-means clustering of genes by including signal intensity of CpG, which sub-divided the previous C3 genes into the C3 (with CGIs) and C4 (without CGIs) as shown in Fig. 2b. This representation revealed not all PCGF1/RING1B-unbound genes were decorated by H3K27ac. Accordingly, we modified the text in lines 235-237.

4) It is important to discuss that PHC2 occupies the 481 genes of cluster1. This indicates a cross-talk between cPRC1 and ncPRC1 at those genes. For this, the authors should investigate if deletion of cPRC1 subunits would also trigger a de-repression of cluster1's genes.

Our reply)

We appreciate these comments. This issue was also asked by the reviewer #1 and was experimentally addressed. To investigate the role of cPRC1 to downregulate the C1 genes in IdHPCs, we have generated *Pcgf2/4*-dKO IdHPCs and performed transcriptomic analysis. Intriguingly, in *Pcgf2/4*-dKO IdHPCs, we did not find up-regulation of the C1 genes, suggesting cPRC1 at C1 promoters is dispensable for their down-regulation. These results are described in lines 255-272 of the main text of the revised version.

5) Does deletion of Ring1A/B also affect expression of myeloid and stemness genes?

Our reply)

We appreciate this comment. Yes, deletion of *Ring1a/b* also resulted in upregulation of myeloid and stemness genes. These results are shown in the figures for the reviewer #2.

6) Treatment with EZH1/2 inhibitor phenocopies the deletion of Pcgf1 without affecting the H2Aub levels. Since the cross talk between ncPRC1 and PRC2 is mediated by H2Aub, it is unclear to the referee which is shared mechanism between these two sets of experiments.

Our reply)

We appreciate this comment. We are sorry for our misleading statement for EZH1/2 inhibitor experiments. What we wanted to show here was a critical role of H3K27me3 to down-regulate the C1 genes in IdHPCs as described in lines 313-315 although cPRC1 incorporating PCGF2/4 was dispensable. We further also noticed that down-regulation of H3K27me3 level and up-regulation of C1 genes expression in *Pcgf1*-KO and *Bcor*^{ΔE9-10} IdHPCs were only modest in comparison with those in UNC1999-treated or *Ring1a/b*-dKO IdHPCs. Based on these observations, we speculated that PCGF1-PRC1 and other variant PRC1 sub-complexes likely compensate each other to mediate H3K27me3 marks at C1 genes in IdHPCs as described in lines 315-319. We further argued how variant PRC1 and PRC2 are linked as described in lines 476-494.

7) Does overexpression of Hmg2a recapitulates Pcgf1 KO?

Our reply)

We appreciate this insightful comment. It is reported that overexpression of Hmga2 leads to the development of myeloproliferative disease with expansion of myeloid biased progenitor cells (Oncogene. 2021 Feb;40(8):1531-1541. doi: 10.1038/s41388-020-01629-w.).

8) In supplementary figure 5a the authors should include Western blot analysis for PCLs proteins.

Our reply)

We appreciate this suggestion. Because immunoblot analysis for PCLs proteins was somehow difficult in IdHPCs, we instead used ChIP-seq analysis for PCL2 as well as JARID2. We found local enrichment of PCL2 and JARID2 was not altered by PCGF1 depletion (Fig. 3a, supplementary Fig. 5c), indicating both PRC2.1 and PRC2.2 are normally recruited to target genes in *Pcgf1*-KO IdHPCs. Accordingly, we described these results in lines 278, 281 and 369-371.

9) In order to claim a direct link between PCGF1 and replication machinery, the authors should perform ColP experiments (in the presence of benzonase), eventually by adding and endogenous tag to PCGF1 for proper IP.

Our reply)

We appreciate this important technical suggestion. According to Reviewer #2's suggestion, we have generated a new allele for *Pcgf1*, which allow the expression of TY1-tagged PCGF1 (Extended Data Fig. 2), and established IdHPCs expressing endogenously TY1-tagged PCGF1. We examined the association of TY1-tagged PCGF1 with PCNA and SMARCA4 by IP and immunoblot analysis in the absence or presence of Benzonase as shown in supplementary Fig. 5h. We observed interaction between PCGF1 and PCNA or SMARCA4 were not disrupted by benzonase treatment, suggesting physical association of PCGF1 with replication machinery (Supplementary Fig. 5h). These results are described in line 223-227, 385-386 and 411-414.

10) It would be interesting to explore the cluster 2, which although is occupied by PRC1.1, does not contain PRC2, changes nucleosome density at TSS upon PCGF1 KO, but seems not to be affected in terms of gene expression.

Our reply)

We appreciate this comment. Similar comments were also given by the reviewer #3. As indicated by the reviewer #2, although the cluster 2 (C2) genes attract PCGF1-PRC1, they failed to attract PRC2 and H3K27me3 instead were decorated by H3K27ac and bound by RNA polymerase 2 (RNAP2). Consistently, the C2 genes were more actively transcribed than the C1 genes but the expression of the C2 genes were barely affected by PCGF1 depletion in IdHPCs

and LMPPs. Therefore, PCGF1 does not contribute to activation per se of the C2 genes. Intriguingly, C2 genes tended to harbor binding motives for transcription factors (TFs) important for B cell differentiation such as PAX2-binding sequence (likely shared by PAX5) and E-box (Supplementary fig. 5l) and, indeed, were bound by PAX5 in proB cells and E2A, an essential TF for B cell development, in HPCs as revealed by ChIP-seq data in the public domain (supplementary Fig. 5m). This may imply that binding of such TFs facilitates the expression of the C2 genes in stage- and/or tissue-specific manner and, thereby, restrain access of PRC2 to target CGIs (ref. 61). In other words, down-regulation of stage- and/or tissue-specific TFs upon differentiation may allow PRC2 recruitment, which facilitate robust downregulation of target genes. As PCGF1-PRC1 is shown to facilitate PRC2 recruitment (ref. 26), we speculate that constitutive binding of PCGF1-PRC1 to the C2 genes contributes for their timely downregulation by receiving developmental signals and to promote differentiation of HSPCs. Consistent with this speculation, we found the C2 genes tended to be down-regulated upon myeloid-skewed differentiation (supplementary Fig. 5n). We argued these points in lines 505-518.

Reviewer #3 (Remarks to the Author):

Manuscript by Takano et al entitled "PCGF1-PRC1 safeguards recovery of nucleosome positioning during S-phase to ensure early hematopoiesis". This is interesting work in which the authors uncover a novel role for the non-canonical PRC1 complex. The authors generate a conditional *Pcgf1* KO mouse model to study the role of PCGF1-PRC1 during early hematopoiesis and find that loss of *Pcgf1* results in myeloid skewing and suppresses lymphopoiesis. Cre-ERT2:*Pcgf1*^{fl/fl} BM cells are transplanted into wt recipient mice and tamoxifen-induced knockout of PCGF11 resulted in reduced BM and spleen cell numbers, a loss of B cell populations but an increase in GMP cells. HSC numbers did not change, but PCGF11 loss results in a change in lineage commitment towards myelopoiesis. These data were independently confirmed by scRNaseq. PCGF1 binds and downregulates myeloid genes, although loss of PCGF1 does not result in loss of PRC2 binding to these myeloid loci. Next, using a proteomics approach, the authors show that PCGF1 interacts with several proteins associated with the replication machinery, and the PCGF1 localizes at the replication fork where it prevents excessive loading of transcriptional activators on nascent DNA. The authors identify that PCGF1-PRC1 facilitates nucleosome deposition immediately after the passage of replication fork, which is required for inheritance of proper chromatin conformation followed by DNA replication. The model that emerges is that at c1 loci

that have H3K27me3, PCGF1 is needed to prevent overloading of chromatin remodelers. In the absence of PCGF1, this does happen, and as a consequence PRC2 activity is reduced and repressive H3K27me3 marks are lost. This occurs particularly at myeloid genes, and therefore after loss of PCGF1 cells undergo myeloid skewing. The authors focus mostly on c1 loci, but these are in fact a minority since c2 loci are more abundant, and since these loci are not in a repressed state (no repressive H3K27me3 marks, but do carry RING1B, H3K27ac, RNAPII) one wonders whether at those loci PcGF1/non canonical PRC1 would fulfill similar functions, most likely not. These are intriguing findings, but I do have a number of comments.

1. "Therefore, the interplay between PCGF1-PRC1 and cPRC1 observed in ESCs does not take place during normal or pathological hematopoiesis, indicating the presence of yet unknown mechanisms by which PCGF1-PRC1 mediates gene repression." It remains unclear why the role of PCGF1-PRC1 would be different in ES cells compared to HSCs, also at the mechanistic level. While in ES cells there appears to be a strong overlap between PRC2 and PRC1 occupied loci that often also carry H3K27me3, in HSCs and LSCs this is clearly different. The authors themselves also show that the majority of PCGF1-bound loci are in fact in an active chromatin conformation (fig.2, c2 loci, 815 loci bound by PCGF11, RING1B, H2K27ac, RNAPII) while less loci are in a repressed conformation (481 c1 loci, PCGF1, PRC2, H3K27me3). Yet, the majority of the story focuses on the role of PCGF1 at these c1 loci, whereby loss of PCGF1 results in overloading of chromatin remodelers at the replication fork, which in turn negatively impacts on PRC2 activity and as a consequence a reduction in repressive H3K27me3 marks is seen, resulting in enhanced myeloid gene expression. The function of PCGF1 at the active loci is then different since there is no PRC2 at all which activity therefore also does not need to be controlled by preventing overloading of chromatin remodelers. This is not addressed.

Our reply)

We appreciate these thoughtful comments. We share reviewer #3's first question why and how PCGF1 functions in distinct manner between ESCs and HPCs. We discussed this point in lines 476-494 in the revised manuscript. Briefly, we guess such difference could come from difference of cellular properties. ESCs are tightly captured and undergo repetitive proliferation without changing cellular properties, in which differentiation program should be robustly inactivated by PcG repressive pathway (ref. 46). Here, PCGF1-PRC1 contributes to initiate PcG-mediated repression by primarily giving H2AK119ub1 (refs. 6, 25), which subsequently recruit PRC2 and canonical PRC1 to build robust repressive domains. In contrast, although

HPCs are mitotically active like ESCs, they are not strictly captured and gradually differentiate into progenitors of respective lineages in proliferation-coupled manner. We therefore speculate canonical PcG repressive pathway that enables robust suppression of the differentiation program as seen in ESCs may not be suitable for HPCs, in which the differentiation program should be executed while keeping their multipotency to some extents. In other words, our results indicate that replication-coupled mechanisms could be competent for the execution of the differentiation programs.

The second comment given by the reviewer #3 is overlapped with that from the reviewer #2 and is discussed in lines 505-518 in the revised version. Briefly, although the cluster 2 (C2) genes attract PCGF1-PRC1, they failed to attract PRC2 and H3K27me3 instead were decorated by H3K27ac and bound by RNA polymerase 2 (RNAP2). Consistently, the C2 genes were more actively transcribed than the C1 genes but the expression of the C2 genes were barely affected by PCGF1 depletion in IdHPCs and LMPPs. Therefore, impacts of PCGF1 to either activation or inactivation are limited at the C2 genes in IdHPCs and LMPPs. Intriguingly, C2 genes tended to harbor binding motives for transcription factors (TFs) important for B cell differentiation such as PAX2-binding sequence (likely shared by PAX5) and E-box (Supplementary fig. 5l) and, indeed, were bound by PAX5 in proB cells and E2A, an essential TF for B cell development, in HPCs as revealed by ChIP-seq data in the public domain (supplementary Fig. 5m). This may imply that binding of such TFs facilitates the expression of the C2 genes in stage- and/or tissue-specific manner and, thereby, restrain access of PRC2 to target CGIs (ref. 61). In other words, down-regulation of stage- and/or tissue-specific TFs upon differentiation may allow PRC2 recruitment, which facilitates robust downregulation of target genes. As PCGF1-PRC1 is shown to facilitate PRC2 recruitment (ref. 26), we speculate that constitutive binding of PCGF1-PRC1 to the C2 genes contributes for their timely downregulation by receiving developmental signals and to promote differentiation of HSPCs. Consistent with this idea, the C2 genes were revealed to tend to be down-regulated upon myeloid-skewed differentiation (supplementary Fig. 5n).

2. The authors appear to ignore somewhat the potential role of non-canonical PRC1 at active loci, not only by focusing mostly on the repressed c1 loci but also in their discussion of the available literature. The authors also exclusively focus on its tumor suppressive roles while various papers have shown that non-canonical PRC1/KDM2B can also act as an oncogene (Andricovich et al, 2016; He et al, 2011; Kottakis et al, 2014; Ueda et al, 2015; van den Boom et al., 2016). These papers highlight a different role and should be discussed.

Our reply)

We appreciate these comments. Potential roles of PCGF1-PRC1 are also argued in our reply to the first comment made by the reviewer #3 as you see in above. We also modified the text in lines 468-472 according to this suggestion.

3. Suppl Fig.1D: why is PCGF1 expression not completely gone? This reduction seems rather modest.

Our reply)

We appreciate this comment. We have replaced the data with those obtained from LMPPs instead of HSCs. In LMPPs, *Pcgfl* transcripts could not be detected as shown in supplementary Fig.1d. This is described in lines 151-153.

4. Figure 1g: why could C3, C6, C7, and C8 clusters not be annotated? Was there a problem with sequence depth? How many transcript were in fact quantified? In the pseudo-time differentiation trajectories depicted in fig 1i: why are the TMPs localized further away from the root state/HSCs compared to GMPs? One would expect that TMPs would precede GMPs? Also MPPs locate in a separate branch, why?

Our reply)

We appreciate these comments. First, as for sequence depth, we could identify 17750 genes (median 4176 genes per cell) in control and 17600 genes (median 2059 genes per cell) in *Pcgfl*-KO. Based on these data, we estimated the depth was appropriate. Instead, annotation was limited by reference data used to annotate developmental stages of respective clusters. We compared our data with publicly available single cell RNA-Seq data, in which representative gene expression profiles for HSCs, MPPs, LMPPs, and GMPs were given (Supplementary Fig. 1h)(ref. 43). As shown in Figs 1e and 1f, C3, C6, and C7 barely exist in the control and are preferentially seen in *Pcgfl*-KO. We therefore could not annotate these clusters based on the comparison with reference data and arbitrarily referred the C3, C6 and C7 as “Transitional Myeloid Progenitors (TMPs)”, which linked HSCs and/or LMPPs with GMP-like in parallel with MPP in *Pcgfl*-KO. In contrast, we identified the C8 as a cluster of B cell progenitors, which are not included in the public dataset used for the cell-type annotation. We therefore referred the C8 as “B-primed”.

We also appreciate the question about of positioning of TMPs and GMPs in the trajectory. It is generally thought to predict the precise order of differentiation based solely (without other given knowledges) on this pseudo-time analysis because, in Monocle, the branches which stem from the same “Spine” are regarded as in equal states. More precisely, Monocle determines a minimum network which comprises all the cells and yields a ”Spine” which is the longest pathway on this network. Then Monocle order cells along with this Spine and reconstructs

branches corresponding to cellular decisions, based on gene expression patterns. During this process, these branches are not in particular order. As branches of TMP and GMP are from the same central root (Fig. 1g), therefore we think it is difficult to discriminate the order of TMPs and GMPs based solely on this analysis. In addition, although some TMPs seem to spread toward the left side, many cells in C3 (TMPs) and C0 (GMPs) are mutually overlapped at the base of the branch (Fig1. g), which could make discrimination of the order of C3 (TMPs) and C0 (GMPs) based on this analysis further difficult.

In regard to distribution of MPPs on the trajectory, our results clearly demonstrate HSCs are allocated to right lower position, MPP/LMPP right upper, GMP left lower, and TMP left upper. This could reflect the difference of self-renewal capacity (right upper vs right lower) and myeloid potential (left vs right) and this could be the reason why MPPs/LMPPs locate in a separate branch.

5. The use of ID3 overexpressing immortalized multipotent progenitor lines is interesting. The authors did not use the inducible model they published previously. Are the authors sure that ID3 does not interfere with non-canonical PRC1 chromatin binding characteristics? In figure 4 primary LMPP or LSK cells were isolated from BM of ERT2-Cre (control) and ERT2-Cre:Pgcf1fl/fl (Pgcf1-KO) mice that were cultured for 4 days in the presence of 4-OHT. These do not express ID3, which would serve as a good control. What I do not understand is why the authors only focus on the 481 c1 loci and 815 c2 loci that were identified in the IdHPCs, why not simply show which genes were effected in these primary HSPCs and then overlay those with what was seen in IdHPCs? Please provide these comparisons as well.

Our reply)

We appreciate this comment. To address this comment, we compared differentially expressed genes (DEGs) in *Pcgf1* deficient LMPPs and observed their expression change in IdHPCs. The results are shown in supplementary Fig. 4d. This analysis revealed that up-regulated DEGs in LMPPs tended to be identified as up-regulated DEGs in IdHPCs (61%: 549 out of 893 genes). This trend was more obvious in the C1 genes (73%: 54 out of 74 genes). Consistently, we did not find considerable difference for distribution of RING1B and H3K27me3 between LMPPs and IdHPCs (Fig. 2b, Fig. 4d). These findings indicated that PCGF1-PRC1 serves as a repressor at C1 genes both in primary LMPPs and IdHPCs, excluding a potential interference for PRC1 by ID3. These results are described in lines 326-328 and 333-335.

6. The effects of *Pcgf1* KO on gene expression changes seems relatively small, even in the c1 subgroup of repressed loci. Are all genes depicted in suppl fig 1e with a log2

FC>1.5 also statistically significant? Same for genes plotted in fig 2e. How many genes are upregulated in suppl fig 1e, and what is the overlap with the c1-c2-c3 subsets in fig 2a-b? Similar for the total up/down genes depicted in suppl fig 2d. Please validate these gene expression changes by independent Q-PCRs/Westerns. HMGA2 is picked as a candidate for further validations, but only changes of chromatin marks are shown, not the effects on gene expression changes, only fig 4g showing genome browser tracks of the expression of Hmga2 in control and Pcgf1-KO LMPPs, which only appears to show modest effects. And is only 1 exon shown? Please clarify. The effects of the by the EZH1/2 inhibitor UNC1999 are clear on HMGA2 expression (Fig. 3f), but this is not the same as PCGF1 KO.

Cluster 2 not only contains B-cell genes, but the strongest group is embryonic organ development (Fig. 2d). What is the function of PCGF1 at these loci? It is very clear that 2 distinct non-canonical PRC1 chromatin states exist in HSPCs, repressed and active states. It remains puzzling what mechanisms control these 2 different states, and what the role of PCGF1 at the active loci is. The authors focus exclusively on the role of PCGF1 at repressed loci and do not discuss at all the potential role of ncPRC1 at active foci.

Our reply)

We appreciate these comments. For the first comment, we experimentally addressed these comments and modified supplementary Fig. 2g and 2h to highlight DEGs with FDR<0.05 as red dots and indicate numbers DEGs. We also performed RT-qPCR for *Hmga2* and *Rxra* and confirmed their up-regulation in *Pcgf1*-KO IdHPCs as shown in Fig. 2g. Up-regulation of HMGA2 was further confirmed by IB analysis as shown in Fig. 2h. We further tested the expression of *Hmga2* and *Rxra* in *Pcgf1*-KO LMPPs and revealed their significant up-regulation as shown in Fig. 4c. We also agree with the Reviewer #3 it is important to explain why down-regulation of H3K27me3 level and up-regulation of C1 genes expression in *Pcgf1*-KO and *Bcor*^{ΔE9-10} IdHPCs were only modest in comparison with those in UNC1999-treated or *Ring1a/b*-dKO IdHPCs. We speculate that PCGF1-PRC1 and other variant PRC1 sub-complexes likely compensate each other to mediate H3K27me3 marks at C1 genes in IdHPCs. We further discussed this issue in lines 315-319. Moreover, we argued how variant PRC1 and PRC2 are functionally linked in lines 476-494.

Second question is about potential roles for PCGF1 in the C2 genes. Similar issue is argued in our reply to the comment 1 given by the reviewer #3. Please refer to lines 495-518.

7. Is it not surprising that the shHmga2-1 with a rather modest 50% knockdown efficiency (Suppl fig. 4c) almost completely restores B cell numbers/%s (fig. 4 j-k)?

Although I do agree that it is of interest to see that overexpression of Hma2 in TET2 deficient background appears to be sufficient to drive myeloid transformation (Bai et al 2021).

Our reply)

We appreciate this comment. To show results in more precise manner, we provide a panel showing B cell numbers in Fig. 4j. It is obvious that sh*Hmga2-1* restored B cell defects in *Pcgfl*-KO but not “completely”. Intriguingly, this result also indicated the importance for fine tuning of *Hmga2* expression in the differentiation of HPCs toward B-cell lineage.

8. Whether the enzymatic activity of PRC2 is truly effected by overloading of chromatin remodelers is formally not shown, the authors refer to 2 papers (refs 22-23) but do not provide data in their model systems in the absence of PCGF1.

Our reply)

We appreciate this critique. To experimentally address this issue, we degraded SMARCA4 by a PROTAC ACB11 in *Pcgfl*-KO IdHPCs (Supplementary Fig. 5i, 5j) and investigated impacts of SMARCA4 degradation on nucleosome density, local H3K27me3 depositions at the C1 genes and their expression. We indeed observed SMARCA4 degradation partially restored nucleosome density on nascent DNA at the C1 genes and concurrently H3K27me3 deposition at *Hmga2* and *Rxra* (Fig. 5g, 5h, 5i). Partial restoration of H3K27me3 deposition at *Hmga2* was further shown to accompany their down-regulation in *Pcgfl*-KO IdHPCs (Fig. 5j). As enzymatic activity of PRC2 is known to be influenced by nucleosome density, these observations support the role of PCGF1 to affect PRC2 activity via regulation of nucleosome density by antagonizing SMARCA4 in replication-coupled manner (ref. 38). These results are described in lines 415-438.

9. Although experimentally probably difficult to address: is the interactome between PCGF1 and DNA replication machinery mostly derived from PCGF1 at c1 repressed loci? In fact the majority of PCGF1 peaks do not appear to be in the repressed c1 loci, and therefore the interactome would not only reflect the situation at c1 loci, but also at active c2 loci. One would presume that loss of PCGF1 also results in an overloading of chromatin remodelers at active c2 loci, what would be the consequence of that? Or rather, why would there not be consequences for those loci, the authors only show that expression of these loci does not appear to altered (eg further increased over basal levels that are already higher). It would be interesting to have the authors thoughts on this.

Our reply)

We appreciate this thoughtful comment. To address this issue, we combined iPOND with ChIP-seq for PCGF1 and found both C1 and C2 genes were bound by PCGF1 even on nascent DNAs, suggesting that PCGF1 associates with C1 and C2 genes during passage of the replication fork (Fig. 5c). Consistent with this, nucleosome density on nascent DNA at the C2 genes was reduced to similar extents to the C1 genes (Fig. 5g, 5h). These results are described in lines 390-396 and 415-438. This reduction of nucleosome density was revealed to be restored by inhibiting SMARCA4 in both C1 and C2 genes. As C2 genes basically lacked PRC2 binding likely due to active transcription, PRC2 cannot be a target for inhibition by SMARCA4 at the C2 genes. Moreover, we revealed that transcriptional status of C2 genes was unaltered in *Pcgl1*-KO IdHPCs. Nucleosome remodeling due to PCGF1 depletion is speculated to minimally affect transcriptional activity of the C2 genes likely due to constitutive association by transcriptional machinery (Fig. 2c).

Minor:

Line 100: "...PCGF4 facilitates maintenance of leukemic stem cells..." please also cite papers focusing on human leukemias (DOI: 10.1182/blood-2009-03-209734; doi.org/10.1182/blood-2010-02-270660)

Our reply)

We thank Reviewer #3 for this suggestion. We have modified the text in lines 113-115 and reference (ref. 29).

Typo in line 195 These cells, therefore, can be utilized as a substitute for LMPPs for.

Our reply)

We thank Reviewer#3. We have corrected it.

Line 198 "We found that PCGF1 was efficiently depleted in IdHPCs after 4 days of 4-OHT injection (Supplementary Fig. 1I)" Injection suggests in vivo but assume 4OHT was administered in vitro?

Our reply)

We appreciate this comment. We have changed "injection" to "treatment".

The +/- 10kb from TSS indicator at the bottom is mispositioned.

Our reply)

We appreciate this comment. We have amended the position of the indicator.

C3=20666 genes, in the VENN diagram in fig.2A it is not immediately clear where this number comes from.

Our reply)

C3 in the previous version represent genes unbound by RING1B. In the revised version, this cluster was further divided into C3 (with CGIs) or C4 (without CGIs) in Fig. 2b. In the Venn diagram in Fig. 2a, genes unbound by RING1B is not shown.

Maybe add fold changes to Suppl Fig 2G, fig.2C, Fig.3B for clarity

Our reply)

We appreciate this comment. We've added fold changes to respective figures for clarify (Supplementary Fig. 2l , Supplementary Fig. 3c, d , Supplementary Fig. 4).

In the discussion, paragraph starting at line 404, please include previous work in the discussion that showed the existence of active loci bound/controlled by non-canonical PRC1 (DOI: 10.1016/j.celrep.2015.12.034)

Our reply)

We appreciate this suggestion. We have modified the text in lines 496-497 and reference (ref. 60).

REVIEWERS' COMMENTS

Reviewer #1 (Remarks to the Author):

The authors have done an excellent job at responding to my previous concerns. In particular, they have added numerous, additional experiments, including studies using endogenously-tagged PCGF1 and many new ChIP studies. Overall, the work is greatly improved, and this should be a valuable study. I had only one minor comment that the authors should address: SMARCA4 is a gene name, whereas Brg1 is the protein name. This should be corrected throughout.

Reviewer #2 (Remarks to the Author):

The authors have addressed by and large my previous concerns.

Reviewer #3 (Remarks to the Author):

my comments have been addressed